# Taste sensing and sugar detection mechanisms in *Drosophila* larval primary taste center

G Larisa Maier[1†], Nikita Komarov[1†], Felix Meyenhofer[1], Jae Young Kwon[2], Simon G Sprecher[1]*

[1]Department of Biology, University of Fribourg, Fribourg, Switzerland; [2]Department of Biological Sciences, Sungkyunkwan University, Suwon, Republic of Korea

**Abstract** Despite the small number of gustatory sense neurons, *Drosophila* larvae are able to sense a wide range of chemicals. Although evidence for taste multimodality has been provided in single neurons, an overview of gustatory responses at the periphery is missing and hereby we explore whole-organ calcium imaging of the external taste center. We find that neurons can be activated by different combinations of taste modalities, including opposite hedonic valence and identify distinct temporal dynamics of response. Although sweet sensing has not been fully characterized so far in the external larval gustatory organ, we recorded responses elicited by sugar. Previous findings established that larval sugar sensing relies on the Gr43a pharyngeal receptor, but the question remains if external neurons contribute to this taste. Here, we postulate that external and internal gustation use distinct and complementary mechanisms in sugar sensing and we identify external sucrose sensing neurons.

## Editor's evaluation

This paper provides an alternative theory about taste coding in larval insects. The authors use a volumetric imaging method to capture neural activity from an entire taste sensory organ in *Drosophila* larvae. Their results suggest that taste coding in *Drosophila* larvae might be different than adult flies, and use multimodal neurons and neural dynamics to represent taste information in the larval brain. Understanding taste coding in *Drosophila* larvae might provide further understanding to how food is encoded in the larval insect brain.

**\*For correspondence:**
simon.sprecher@unifr.ch

†These authors contributed equally to this work

**Competing interest:** The authors declare that no competing interests exist.

## Introduction

The sense of taste provides an innate ability among animals to avoid toxic or deleterious food and to choose a nutritious diet. Despite comprising only few chemosensory neurons at the periphery (*Stocker, 1994*; *Stocker, 2008*), *Drosophila* larvae are able to sense a wide range of chemicals but the principles of how a small neuronal population encodes a large sensory space entail unresolved questions (*Komarov and Sprecher, 2022*). They display behaviors to different gustatory cues (*Gerber and Stocker, 2007*), being repelled by bitter tastants (*El-Keredy et al., 2012*; *Kim et al., 2016*) as possible signals of toxic food, and attracted by sweet, certain amino acids (*Schipanski et al., 2008*; *Croset et al., 2016*; *Kudow et al., 2017*), or ribonucleosides (*Mishra et al., 2018*), indicators of a nutritious diet. For *Drosophila* larvae, an animal that increases body size up to 200–250-fold in 5 days (*Robertson, 2009*) and spends the bulk of its time searching for food (*Green et al., 1983*), the most important nutritional sources are sugar, such as sucrose and fructose found in fermenting fruits – their natural diet, as well as proteins and fatty acids, taken up from yeast

growing on this substrate. As one of the most important macronutrients in the larval diet, sugar is associated with high nutritional values for survival and with proper function of metabolic processes. Nevertheless, larval external taste neurons have not been associated thus far to a role in sugar taste preference.

In the fruit fly, taste encoding at the periphery has been described in line with segregated hedonic valence logic, attractive and aversive taste being detected through separate cell populations. Accordingly, specific members of the *Gustatory receptors* (*GRs*) family mediate sweet or bitter taste, namely cells expressing Gr5a, Gr61a, or Gr64a-f detect sugars and do not overlap with bitter-taste cells expressing *Gr66a* or *Gr33a* (*Dahanukar et al., 2007*; *Fujii et al., 2015*; *Thorne et al., 2004*; *Liman et al., 2014*; *French et al., 2015*). Additional taste modalities such as salt, fatty acid, carbonation, polyamines, or amino acids partially overlap onto these by means of added-on expression of *Ionotropic receptors* (*IRs*) or *pickpocket* receptors (*Ppk*) (*Zhang et al., 2013*; *Hussain et al., 2016*; *Lee et al., 2017*; *Tauber et al., 2017*; *Ahn et al., 2017*; *Ganguly et al., 2017*; *Sánchez-Alcañiz et al., 2018*; *Jaeger et al., 2018*), but maintaining a segregated attraction/aversion quality of taste in the separate cell populations. Contrary to described adult fly taste sensing logic, in *Drosophila* larva taste of opposed valence can overlap, shown by a single gustatory receptor neuron (GRN) activated by both sugar and bitter tastants (*van Giesen et al., 2016a*) and thereby indicating a higher level of taste integration than in the adult fly.

Chemosensory organs are located at the tip of larval head, and comprise bipolar sensory neurons that extend axonal projections into defined regions of the brain (*Figure 1A*; *Python and Stocker, 2002*). The main external sense organs are the terminal organ ganglion (TOG) with taste primary function and the dorsal organ ganglion (DOG) as olfactory organ (*Stocker, 1994*; *Klein et al., 2015*). Out of more than 30 GRNs in the TOG only seven neuronal identities (C1–C7) have been mapped and are traceable by means of individual GR-Gal4 lines (*Figure 1B*; *van Giesen et al., 2016a*; *Kwon et al., 2011*; *Rist and Thum, 2017*). Because of experimental restrictions in probing the system due to the limited availability of individually mapped TOG Gal4 lines, physiological characterization has been thus far tackled in few single GRNs, but a global approach for measuring responses in larval taste sense neurons could prove useful to an overview of peripheral gustatory responses.

Therewithal, larval repertoire of taste receptors differs from expression described in adult flies, with many GRs showing developmental stage-specificity. *Gr66a/Gr33a*-expressing neurons, mediators of bitter perception in the adult fly, are also present in the larva and have been associated with roles in bitter taste avoidance (*Kim et al., 2016*; *Apostolopoulou et al., 2014*; *Choi et al., 2020*). However, sweet sensing neurons or receptors have not been identified in the larval peripheral taste system (*Kwon et al., 2011*), although larvae can detect and elicit behaviors towards most sugars (*Rohwedder et al., 2012*). Larval sugar perception has been allotted to *Gr43a* fructose receptor (*Mishra et al., 2013*), which, however, is expressed in pharyngeal sensory organs, foregut and in the brain, but absent from external taste neurons, similar to its expression in the adult fly where it acts as main hemolymph fructose sensor (*Miyamoto et al., 2012*; *Miyamoto and Amrein, 2014*). As one of the most important macronutrients in larval diet, and therefore it seems surprising that external taste organs at direct contact with animal's environment would not be involved in sugar detection and output behavior, all the more so as sucrose can elicit physiological response in TOG neurons (*van Giesen et al., 2016a*).

In this study, we probe response profiles of TOG taste neurons by means of whole-organ in-vivo calcium imaging and subsequently pursue to investigate sugar peripheral taste in the larva. Our experimental approach allows an assessment of general tuning of larval GRNs and identification of distinct response dynamics, while revealing a high fraction of calcium response to sucrose. We test larval behavior and tackle the described central role of Gr43a receptor in sugar taste by separating external taste sensing from internal taste. Segregation of internal and external larval taste neurons in behavior has also been recently established for bitter taste (*Choi et al., 2020*). We equally explore the contribution of individual TOG neurons in sugar taste and identify C2 as one neuron participating in peripheral sucrose detection in the larva. Our study reveals a high complexity in taste response at the level of external taste neurons in the fruit fly larva, and we propose that distinct neurons are employed in sugar sensing depending on the dietary substrate.

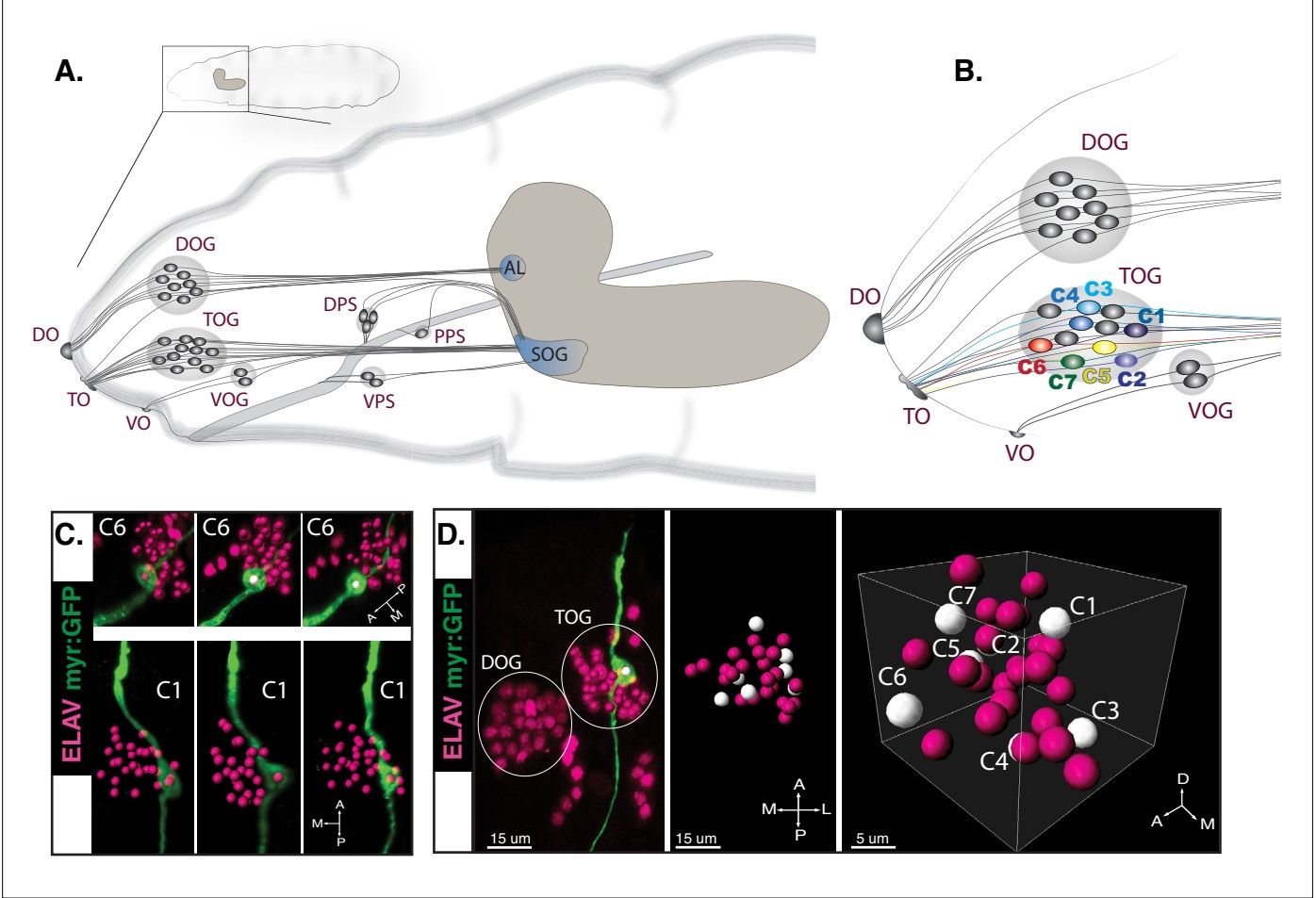

**Figure 1.** Peripheral taste neurons of the fruit fly larva. (**A**) The chemosensory system of the larva. External sense organs extend dendrites to the periphery and have main olfactory function—the DOG (dorsal organ ganglion), or gustatory function—the TOG (terminal organ ganglion) and the VOG (ventral organ ganglion). Dendrites of the internal dorsal, ventral, and posterior pharyngeal sense organs (DPS, VPS, and PPS) innervate the pharynx, thus involved in taste sensing during food ingestion. All chemosensory neurons project axons to the brain in the subesophageal ganglion (SOG)—first central taste integration relay—or to the antennal lobe (AL) for central olfactory processing. (**A**) Has been adapted from (**C**) from *Gerber and Stocker, 2007*. (**B**) External chemosensory organs. Seven GSNs previously identified and named C1–C7 are represented here by the color code they were first described with (*van Giesen et al., 2016a*; *Kwon et al., 2011*). Used Gal4 lines: *Gr22e*-Gal4 (C1), *Gr94a*-Gal4 (C2), *Gr66a*-Gal4 (C1, C2, C3, and C4), *Gr59e*-Gal4 (C5), *Gr21a*-Gal4 (C6), and *GMR57BO4*-Gal4 (C7). (**C**) UAS-*myrGFP* reporter was expressed in individual GSNs using corresponding Gal4 lines (**B**). We observed a relatively stereotypic position of specific neurons within the organ across animals (n≥3)—exemplified on C6 (*Gr21a*-Gal4) and C1 (*Gr22e*-Gal4). (**D**) Illustrative 3D map of TOG segmented cells. Position of neurons with known identities (white dots) is approximated based on separate immunostainings.

The online version of this article includes the following video and figure supplement(s) for figure 1:

**Figure supplement 1.** Whole-organ calcium imaging—data processing.

**Figure supplement 1—source data 1.** Whole organ calcium imaging—data processing.

**Figure 1—video 1.** Whole-organ recording and cell segmentation.

https://elifesciences.org/articles/67844/figures#fig1video1

## Results
### Whole organ larval taste recordings with cellular resolution

At the level of the TOG, main larval external taste organ containing around 30 sense neurons, 7 GRNs have been identified based on receptor-Gal4 expression (*van Giesen et al., 2016a*; *Kwon et al., 2011*). To assess their spatial location, we performed immunostainings of the seven individual TOG neurons (*Figure 1C and D*). Although cell positions vary slightly between stainings (n≥3) (*Figure 1C*),

3D locations can be approximated on an illustrative map (*Figure 1D*). Four individual TOG neurons have previously been characterized physiologically, revealing taste multimodality as a characteristic in the larval sensory system (*van Giesen et al., 2016a*). Considering that molecular mapping for all TOG neurons through single driver lines has not yet been compiled, physiological characterization of the larval external taste organ through labeling individual neurons is not readily achievable. To circumvent this constraint in direction for a response profile overview of larval GRNs in the TOG we implemented a whole-organ imaging approach.

We performed in-vivo whole-organ calcium imaging recordings by expressing cytoplasmic GCaMP6m (*Chen et al., 2013*) and nuclear RFP (*Egger et al., 2007*) reporters in all neurons (*Figure 2A*, *Figure 1—figure supplement 1*, *Figure 1—video 1*). The use of a nuclear reporter was chosen for efficient cellular segmentation and for establishing a reproducible data processing pipeline for whole organ recordings (*Figure 1—figure supplement 1—source data 1*). As previously described (*van Giesen et al., 2016b*), we made use of a customized microfluidic chamber allowing for simultaneous imaging and chemical stimulation. We used a series of chemical compounds belonging to different canonical taste categories in concentrations that have been previously used in similar studies in the fruit fly (*van Giesen et al., 2016a*; *Park and Carlson, 2018*; *Ling et al., 2014*; *Table 1*, *Figure 2*, see Materials and methods).

To start with, we grouped gustatory cues used for GRN stimulation in groups by taste category for sweet, bitter, amino acid, and salt taste modalities (*Table 1*) and pooled all neuronal responses for the ratio of taste modality integration per cell (*Figure 2B and C*). In the sweet taste category, we tested two sugar groups chosen as monosaccharides (fructose, glucose, arabinose, mannose, and galactose) and disaccharides (sucrose, trehalose, maltose, lactose, and cellobiose) and calculated approximately 74% of sugar-responsive neurons to be activated by either monosaccharides (41%) or disaccharides (33%), and approx. 25% by both groups (*Figure 2B*, left panel). For salt taste stimulation, we mixed NaCl and KCl at final concentration of 50 mM for low salt group, and at 1 M for high salt group. We observed a low percentage of cell activation overlap among salt-responding cells with 20% activated by both high and low concentrations (*Figure 2B*, middle panel) and we recorded more responses to high salt (56%) than to low salt. The 20 proteinogenic amino acids were compiled in four groups (A, B, C, and D) as previously described by *Park and Carlson, 2018* (*Table 1*, see Materials and methods). 38% amino acid-responding neurons showed activation to two or more amino acid groups (*Figure 2B*, right panel). Bitter tastants were randomized in two groups of four substances each (*Table 1*, see Materials and methods): DSoTC group (denatonium benzoate, sucrose octaacetate, theophylline, and coumarin), and respectively QLSC group (quinine, lobeline, strychnine, and caffeine) chosen at previously used concentrations (*Ling et al., 2014*). Among cells responsive to bitter stimulation, we observed a bigger fraction of cells activated by DSoTC group (55%). In animals tested for sweet and bitter taste response, we identified cells activated by both sugar and bitter groups (30%, *Figure 2C*, right panel), in agreement with cell integration of opposite valence tastants.

## Sucrose elicits neuronal responses in the TOG

Further, we focused GRN stimulation on few specific compounds representative of the five canonical taste categories: sweet, bitter, salt, amino acids, and sour. Each animal was stimulated with a representative substance from each category, either with sucrose (sweet), denatonium (bitter), NaCl 1 M (high salt), valine (amino acid), and citric acid (sour) or with sucrose (sweet), quinine (bitter), NaCl 100 mM (low salt), arginine (amino acid), and citric acid (sour) (*Figure 2D*). We recorded a 15 total of TOG organs and we selected activation responses calculated above a threshold of 20% DF/F0. Out of all responding neurons (N=197, 15 organs), 68% showed activation to only one tastant per animal and most unimodal responses were recorded to sucrose and to high salt (*Figure 2D'*, left extended pie chart). Up to 32% responding neurons were activated by more than one test substance used for stimulation of the animal. The most frequent taste modalities combinations eliciting response in the same neuron within a given preparation (*Figure 2D'*, right extended pie chart) were: sucrose+ citric acid, sucrose and valine/arginine (sweet+ amino acid, 17%), sucrose and high salt (sweet+ high salt, 19%), sucrose and low salt (10%), sucrose and denatonium/quinine (sweet+ bitter, 9%), sucrose and citric acid (9%), citric acid and high salt (sour+ high salt, 12%), and finally amino acid and low salt (10%). Interestingly, therefore, sucrose was the tastant corresponding to several uni-taste/cell responses but also to most frequent taste combinations that activate the same neuron.

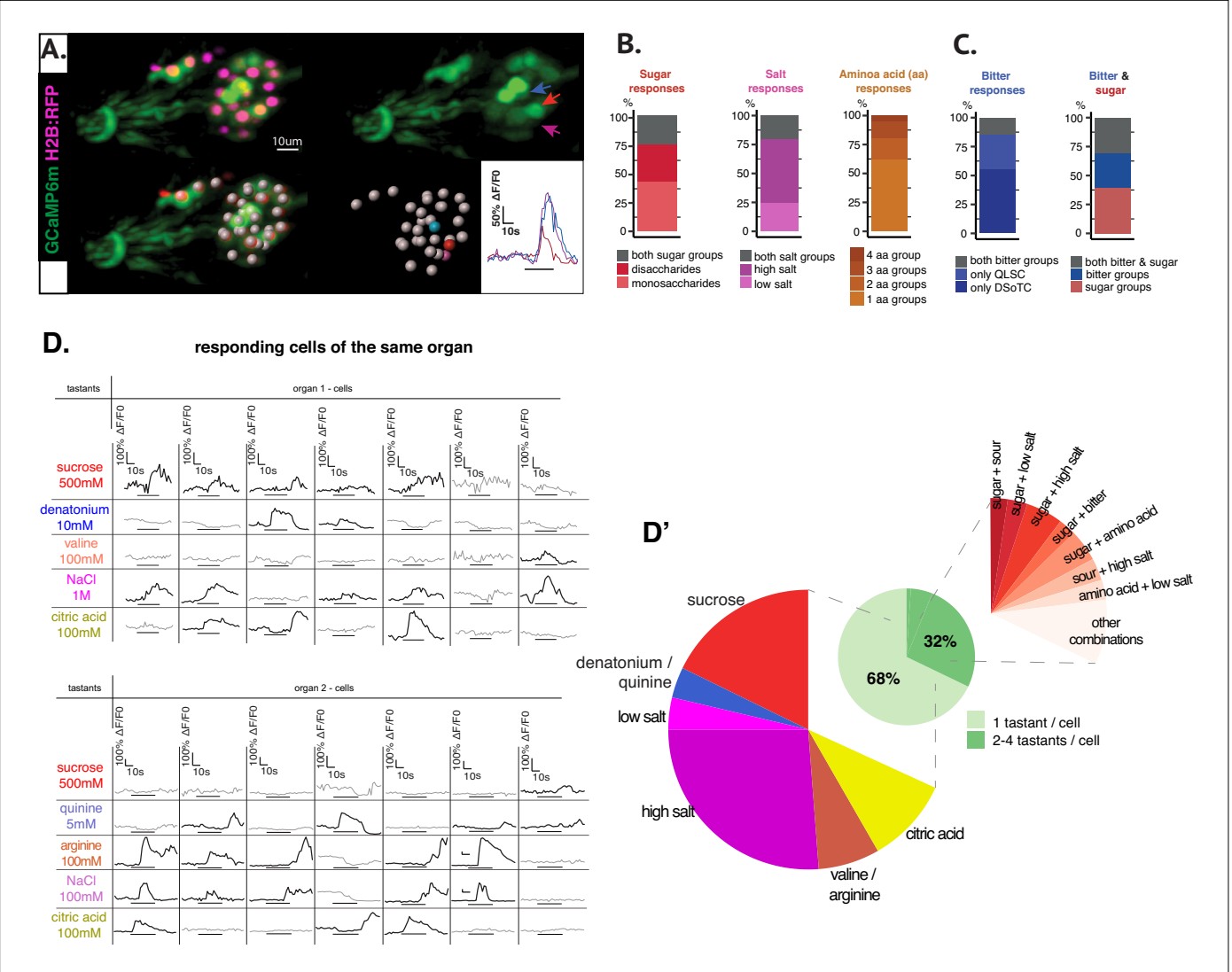

**Figure 2.** Whole-organ calcium imaging shows an overview of response tuning of GRNs. (**A**) Representative recording of the larval primary taste organ stimulated with citric acid 100 mM. Cytoplasmic expression of *GCaMP6m* and nuclear expression of *RFP* in all neurons using *nsyb*-Gal4 driver line (upper left panel) and cell segmentation (white spots, lower left panel). Responding neurons are indicated by blue, red, and magenta arrows (upper right panel) and corresponding dots and fluorescence traces in the same colors (lower right panels). (**B**) Stimulation with groups of tastants by taste modality, in order: two sugar categories (mono and disaccharides), low and high concentration salts, and four amino acid groups. A response intensity threshold of 20% DF/F was considered and neuronal responses across different preparations were pooled together. Among all neurons responding to sugar group stimulation, a comparable ratio of cells showed activation to either one or the other category and 25.4% to both groups of mono and disaccharides. For salt-responding neurons, a higher proportion of neurons showed activation to high salt than to low salt concentration, and 20% neurons responded to both categories. Amino acids split into four groups gave a rounded 62% of uni-group responses and the rest of 38% neurons responded to two or more amino acid groups per animal. (**C**) Responses in animals stimulated with two groups of bitter compounds or with both sugar and bitter test groups. Within the total number of bitter-responsive neurons, 55% were uniquely activated by the DSoTC group and respectively 30% by QLSC group. In animals stimulated with both modality groups, integration between sweet and bitter taste has been calculated and we observed similar proportions of taste cells responding only to bitter groups (30.3%), cells responding only to sugar (39.4%) and cells responding to groups of both modalities (30.3%). (**D**) GRN responses to series of individual tastants. Taste stimulation was done using a series of five tastants per animal, each belonging to one of the five canonical taste modalities represented by sucrose 500 mM (sweet/red color), denatonium 10 mM or quinine 5 mM (bitter/blue), NaCl 1 M for high salt or NaCl 100 mM for low salt (salt/purple), valine 100 mM or arginine 100 mM (amino acids/brown) and citric acid (CiA) 100 mM (sour/yellow). Seven and respectively eight separate organs were recorded within the two series of stimulation (N=15) and a total of 197 neurons showed responses above 20% DF/F0 and were further analyzed. The table shows illustrative traces for cells responding to one or more tastants in two separate representations, each for one of the two stimulation sets. (**D'**) Total percentages of taste integration in pooled data are represented on the pie chart: 68% of total responding neurons were activated by only one tastant per organ and up to 32% of neurons responded to more than one tastant. Most unimodal responses (uni-taste/cell, left extended pie) were recorded to sucrose and to high salt. Conversely, numerous neurons were activated by different combinations of taste

*Figure 2 continued on next page*

*Figure 2 continued*

categories, with different frequency of occurrence (multi-taste/cell, right extended pie). Among the most frequent combinations of taste modalities activating the same cell we noted sucrose (sweet)+ any other taste category. Some of these co-modalities involve a presumed positive valence taste (sucrose/sweet) together with a negative valence taste (bitter or high salt) sensed by the same neuron.

The online version of this article includes the following figure supplement(s) for figure 2:

**Source data 1.** Referring to graphs B & C.

**Source data 2.** Excel sheet one refers to data in *Figure 2D'*, comprising the list of neuronal responses above 20% DF/F0.

**Figure supplement 1.** Whole-organ calcium imaging—an overview of response tuning of GRNs per recorded organ.

**Figure supplement 2.** Single neuron sugar responses based on order.

**Figure supplement 2—source data 1.** Referring to *Figure 2—figure supplement 2*.

The above fractions observed across pooled data recapitulate the responses recorded per each organ, as represented in *Figure 2—figure supplement 1*, with an average of taste modality integration per cell per organ of 67% multimodality/33% unimodality, for a mean of 19 responses per organ, 13 responding neurons, and 5 sucrose-activated neurons per organ detected in average.

To test if order of tastant presentation would generally alter neuronal responses within a series of stimulations, we looked into responses of C7 to sucrose 500 mM before or after the denatonium stimulation. C7 is a TOG neuron previously shown to respond to both sugar and bitter substances (*van Giesen et al., 2016a*), but we observed no significant difference in sucrose-elicited response in this neuron based on order of sweet and bitter presentation (*Figure 2—figure supplement 2*).

## Distinct response temporal dynamics

The observations described above for grouped or individual series of tastant stimulations refer to canonical recorded calcium signals characterized by a rise in GCaMP fluorescence during stimulus application. However, at a closer look, tastants elicit different types of signals within the TOG organ, besides canonical ON response, neurons also showing OFF responses and late activation or even fluorescence intensity drop upon stimulation, as exemplified in *Figure 3—figure supplement 1A* for sucrose 500 mM, citric acid 100 mM, or NaCl 100 mM. We proceeded to manually map responses to organ across different larvae and for this purpose, we compiled all activation signals of early/ON responses or late onset/OFF responses with a fluorescence rise above 20% response threshold. Four neurons localized anterior-laterally to the other cells (named Anterior Lateral Neurons [ALNs], *Figure 3* insets) served as landmark in determining the relative spatial location of a cell. We mapped a tastant to a cell location if it associated a response at the respective identity in more than four organs (n≥4, *Figure 3A*, *Figure 3—figure supplement 1B, C*), except for bitter-evoked responses for which n≥3 (*Figure 3—figure supplement 1B, C*). Mapped cells were named based on their spatial location (e.g., PM1=posterior-medial 1, *Figure 3B*).

In line with unmapped responses summarized in *Figure 2D*, by mapping cellular calcium signals, we found neurons with different tuning profiles and sucrose responses were mapped to different TOG cells (*Figure 3—figure supplement 1C*, *Figure 3—source data 1*).

Tastant-evoked deactivation (fluorescence drop) was identified for different test substances (*Figure 3—figure supplement 1A*) but mapping this type of signals was evident especially in the case of one neuron in the organ named CDL2 (central-dorsal-lateral neuron 2). In particular for citric acid, this neuron was often recorded to respond with a decrease of fluorescence while its neighbor, CDL1, typically showed canonical calcium activation to citric acid (*Figure 3A*, *Figure 3—figure supplement 1B*).

## Sugar sensing in *Drosophila* larva

### TOG neurons can detect and contribute to sucrose preference behavior

Larval sugar taste has been associated with the pharyngeal and brain expression of Gr43a receptor (*Mishra et al., 2013*). Yet, as we describe above and as previously shown (*van Giesen et al., 2016a*), sucrose elicits calcium responses in external chemosensory neurons, raising the question of a potential additional role of TOG neurons in sugar detection, which we set to explore next in this study.

**Table 1.** Chemicals used for taste stimulation in calcium imaging.

| Tastant or taste group | Compound | Associated taste (humans) | Concentration | Group final Concentration |
|---|---|---|---|---|
| | Sucrose | Sweet | 500 mM | – |
| | Denatonium benzoate | Bitter | 10 mM | – |
| | Valine | Amino acid/Bitter | 100 mM | – |
| | NaCl | Salty | 1 M | – |
| Tastant series 1 | Citric acid | Sour | 100 mM | – |
| | Sucrose | Sweet | 500 mM | – |
| | Quinine | Bitter | 5 mM | – |
| | Arginine | Amino acid/Bitter | 100 mM | – |
| | NaCl | Salty | 100 mM | – |
| Tastant series 2 | Citric acid | Sour | 100 mM | – |
| | Fructose | Sweet | 100 mM | |
| | Glucose | Sweet | 100 mM | |
| | Arabinose | Sweet | 100 mM | |
| | Mannose | Sweet | 100 mM | |
| Group sugars Monosaccharides | Galactose | Sweet | 100 mM | 500 mM |
| | Sucrose | Sweet | 100 mM | |
| | Trehalose | Sweet | 100 mM | |
| | Maltose | Sweet | 100 mM | |
| | Lactose | Sweet | 100 mM | |
| Group sugars Disaccharides | Cellobiose | Sweet | 100 mM | 500 mM |
| | Denatonium benz. | Bitter | 1 mM | |
| | Sucrose octaacetate | Bitter | 1 mM | |
| | Theophylline | Bitter | 10 mM | |
| Group bitter DSoTC | Coumarin | Bitter | 10 mM | 22 mM |
| Group bitter QLSC | Quinine | Bitter | 1 mM | 13 mM |
| | Lobeline | Bitter | 1 mM | |
| | Strychnine | Bitter | 1 mM | |
| | Caffeine | Bitter | 10 mM | |

*Table 1 continued*

| Tastant or taste group | Compound | Associated taste (humans) | Concentration | Group final Concentration |
|---|---|---|---|---|
| | Valine | Bitter | 10 mM | |
| | Leucine | Bitter | 10 mM | |
| | Isoleucine | Bitter | 10 mM | |
| | Methionine | Bitter | 10 mM | |
| | Tryptophan | Bitter | 10 mM | |
| Group Amino acids A | Cysteine | Bitter | 10 mM | 60 mM |
| | Alanine | Sweet | 10 mM | |
| | Phenylalanine | Bitter | 10 mM | |
| | Glycine | Sweet | 10 mM | |
| | Proline | Sweet | 10 mM | |
| Group Amino acids B | Tyrosine | Bitter | 1 mM | 41 mM |
| | Arginine | Bitter | 10 mM | |
| | Lysine | Salty-Bitter | 10 mM | |
| | Aspartic acid | Umami | 10 mM | |
| | Glutamic acid | Umami | 10 mM | |
| Group Amino acids C | Histidine | Bitter | 10 mM | 50 mM |
| | Serine | Sweet | 10 mM | |
| | Threonine | Sweet | 10 mM | |
| | Asparagine | Neutral | 10 mM | |
| Group Amino acids D | Glutamine | Sweet | 10 mM | 40 mM |
| | NaCl | Salty | 500 mM | |
| Group high salt | KCl | Salty | 500 mM | 1M |
| | NaCl | Salty | 25 mM | |
| Group low salt | KCl | Salty | 25 mM | 50 mM |

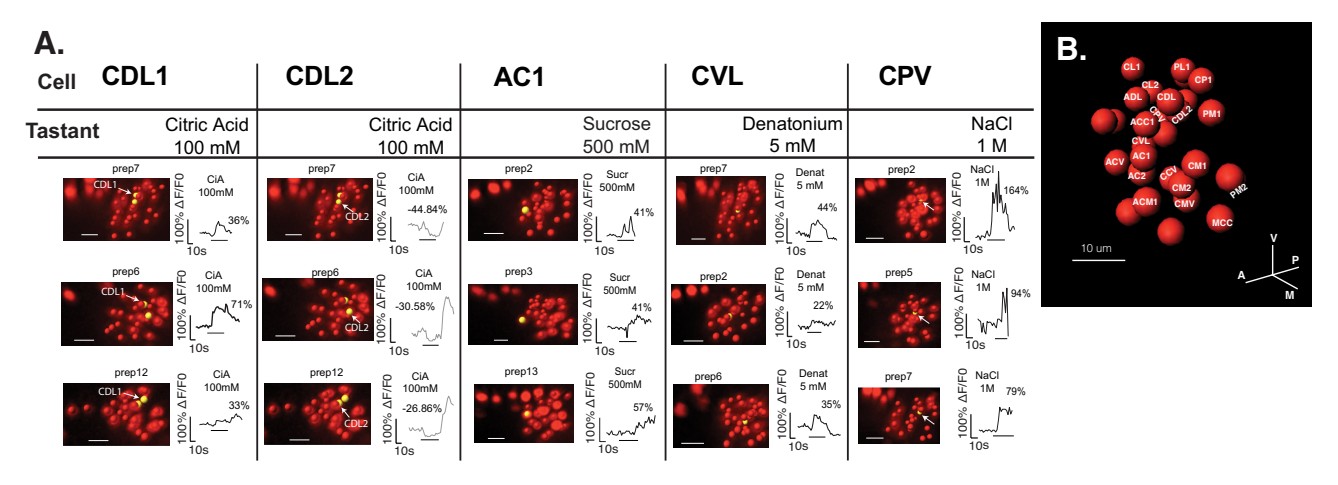

**Figure 3.** Taste-elicited responses mapped to neurons across animals. (**A**) Illustrative traces for a given tastant stimulation from three different organs are shown for manually mapped neurons. (**B**) Representative map used for matching neuronal identities across different recorded organs. Names were chosen as reference for difference in neighboring cell positions, suggested with respect to the sagittal plane, anterior-posterior, and ventral-dorsal axis (C=central/center, V=ventral, D=dorsal, P=posterior, A=anterior, M=medial/n, L=lateral).

The online version of this article includes the following figure supplement(s) for figure 3:

**Source data 1.** Referring to *Figure 3* and *Figure 3—figure supplement 1C*; excel file comprising responses associated to mapped cells.

**Figure supplement 1.** Responses across animals show broad tunning of GRNs and distinct temporal dynamics.

To start with, by means of cell silencing using tetanus toxin reporter UAS-*TNTE* (*Sweeney et al., 1995*), we looked into the role of individual TOG neurons that have previously been shown calcium imaging responses to sucrose, respectively C2, C6, and C7 (*Figure 4*). Behavioral preference of L3, 4-day-old larvae were tested as described before (*van Giesen et al., 2016a*) by way of two-choice test after 5 min on 2.5% agarose petri-dishes (see Materials and methods). Among the three probed neurons, only C2-silencing showed a preference index different from controls (*Figure 4A*, one-way ANOVA p<0.05), while C6 or C7 silencing did not show a notable behavioral defect. We equally tested Gal4 lines yielding expression in several neurons that also include C2 (*Figure 4—figure supplement 1A*) and observed a defect for silencing these lines, although more diminished than for C2 silencing alone and not significantly different from controls. Silencing C5 neuron, TOG neuron that does not express expressed *Gr66a/Gr33a* bitter co-receptors, also showed a trend for a reduced preference (*Figure 4—figure supplement 1A*).

Although it expressed *Gr66a/Gr33a* bitter co-receptors, a possible role of C2 in detecting an attractive tastant is not conflicting with previous observations. Behavioral taste screening studies have not linked C2 with bitter taste detection by contrast with other *Gr66a*-expressing neurons (*El-Keredy et al., 2012*; *Kim et al., 2016*) and, furthermore, optogenetic activation of C2 actually associates a 'weakly attractive' larval response (*Hernandez-Nunez et al., 2015*).

C2 silencing shows a defect in sucrose preference at different concentrations—100 mM, 500 mM, and 1 M, measured at different time points (2, 5, and 15 min *Figure 4B*— upper panels).

Given that *Gr43a* receptor has been described to carry a predominant role in larval sugar detection, we noted agarose concentration differences between our tests and the behavioral assay described by *Mishra et al., 2013*. Our two-choice assays have been developed on 2.5% agarose substrate (see Materials and methods), while Mishra et al. performed tests using 1% concentration. Interestingly, when we tested sucrose preference on 1% substrate, we observed a diminishment of the defect in C2-silenced larvae compared to control (*Figure 4B*, lower panels).

## Larvae can detect sugar with or without food ingestion

To test if the agarose concentration difference reflects a divergent larval capacity to eat, we compared the two substrates by adding sucrose 500 mM and Brilliant Blue 2% and checked for ingested agarose in the larval intestine (*Figure 4C*) at different time points. Excitingly, we detected blue dye in the

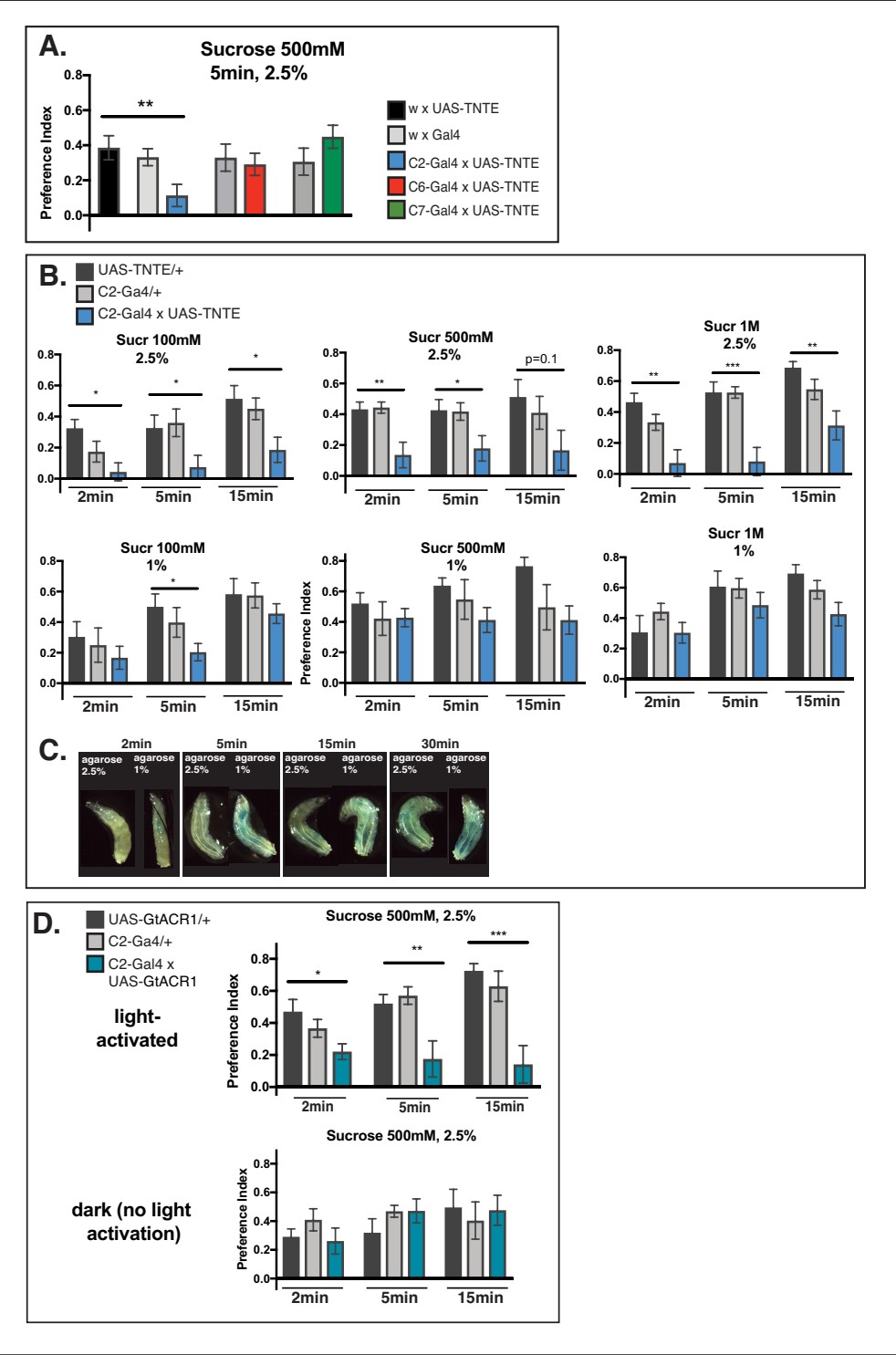

**Figure 4.** Mechanism of larval sugar taste at the periphery. (**A**) Behavioral tests on C2, C6, or C7 neuron silenced larvae. Although all three neurons have previously been associated with physiological responses to sucrose 500 mM (**van Giesen et al., 2016a**), a defect in preference index to Sucrose 500 mM compared to control larvae is found only for C2 (p-value=0.038), indicating a direct contribution of this neuron in sucrose sensing. Driver lines: *Gr94a*-Gal4 (C2), *Gr63a*-Gal4 (C6), and *GMR57BO4*-Gal4 (C7). N=10 for all genotypes; one-way ANOVA; *p<0.05, **p<0.005, ***p<0.001; plot as mean with SEM. (**B**) The role of the C2 neuron in preference to a range of sucrose concentrations on 1% versus 2.5% agarose. Larvae with C2 silenced show a greater defect for sugar preference on 2.5% agarose compared to 1% for all concentrations. This suggests larvae rely at least partly on C2 in sucrose

*Figure 4 continued on next page*

*Figure 4 continued*

sensing on a denser substrate. At 1%, C2 silencing has a lesser effect on sugar preference, suggesting that other mechanisms come into play for behavior on a softer substrate. N=10 for all genotypes; one-way ANOVA, *p<0.05, **p<0.005, ***p<0.001; plot as mean with SEM. (**C**) Food ingestion measurement using blue dye. Larvae were tested at different timepoints to compare eating on agarose 1% versus agarose 2.5%, both containing Sucrose 500 mM. Blue coloring of the intestine was observed as early as 2 min on 1% concentration, while on 2.5% animals didn't seem to be able to eat as easily as indicated by the lack of tint of their abdomen at 2 min and 5 min testing. For longer time points, larvae also start showing food ingestion on 2.5% agarose, but the intestinal blue dye is less prominent than on 1% concentration. This suggests that animals easily eat the softer 1% food substrate but that on denser 2.5% concentration ingestion is slower and evidently impaired especially for early test time points. (**D**) Temporally restricted inhibition of C2 in sucrose 500 mM sensing. Alternative silencing of C2 using *G. theta* anion channelrhodopsin 1 (GtACR1) confirms the observed defect for sugar preference on 2.5% agarose across the timelapse. In dark conditions (no light activation), a defect is not noted. N=7–8; one-way ANOVA, *p<0.05, **p<0.005, ***p<0.001; plot as mean with SEM.

The online version of this article includes the following figure supplement(s) for figure 4:

**Source data 1.** Referring to *Figure 4*.

**Figure supplement 1.** Mechanism of larval sugar taste at the periphery.

**Figure supplement 1—source data 1.** File containing larval preference indexes presented in *Figure 4—figure supplement 1*, as well as corresponding statistics and p-values.

---

animal intestine after as early as 2 min in larvae probed on the 1% agarose substrate but no blue coloration in the abdomen on larvae from 2.5% agarose. At 5 min preference—representative time point we generally employed in two-choice experiments as, for example, in *Figure 4A* —we could equally see the difference in ingested agarose, reflected by more blue dye intake in larvae wandering on 1% substrate compared to larvae on 2.5%. After 15 and respectively 30 min, larvae probed on 2.5% agarose also appeared blue colored, but visibly less than larvae tested on 1% concentration (*Figure 4C*). These observations suggest that on 1% agarose larvae can more readily ingest and therefore likely employ the internal taste organs for navigating the environment, while on a more solid consistency of 2.5% agarose larvae cannot immediately ingest the substrate and probably rely on external chemosensory organs for sampling their food substrate.

## Complementary taste sensing mechanism for larval sugar preference

To test this hypothesis, we assessed the role of Gr43a receptor in the detection of sucrose 100 mM and 500 mM, as well as to fructose 100 mM and 500 mM on both 1% and 2.5% agarose (*Figure 4—figure supplement 1B, B'*). We have employed the same genetics and fly lines as previously described by *Mishra et al., 2013*, testing *Gr43a* mutant, *Gr43a* rescue, and '*Gr43a* brain-only' larvae, which, as described in the referenced study, lack *Gr43a* peripheral expression but keep brain receptor expression intact.

The *Gr43a* mutant showed diminished preference at different time points to both sucrose concentrations on 1% agarose (*Figure 4—figure supplement 1B*, upper panels), whereas on 2.5% mutant preference was more similar to rescue control larvae (*Figure 4—figure supplement 1B*, lower panels). Similar observations could be made for the '*Gr43a* brain-only' larvae tested on sucrose 500 mM that showed a defect at immediate preference on 1% but no defect on 2.5%.

When preference to fructose 100 mM was tested on 1% agarose, *Gr43a* mutant and '*Gr43a* brain-only' showed a completely abolished immediate preference (2 min) (*Figure 4—figure supplement 1B'*, upper left panel), as expected from previously reported results (*Mishra et al., 2013*). Moreover, equally reproducing observations from the referenced study for *Gr43a*, the 'brain-only' larvae gradually rescue the defect, in line with the idea that brain receptors compensate fructose sensing at delayed preference (15 min) (*Mishra et al., 2013*). Interestingly, the mutant defect appears to be lesser at higher fructose concentrations and especially on 2.5% substrate concentration (*Figure 4—figure supplement 1B'*). These observations suggest that larvae deficient of Gr43a receptor in pharyngeal taste neurons rely on other taste sensing mechanisms when unable to eat, such as via the C2 neuron in the TOG, and also that Gr43a in the larva, as in the adult (*Miyamoto et al., 2012*), is more attuned to fructose sensing than to sucrose.

We also tested *Gr43a* brain silenced larvae (*Figure 4—figure supplement 1C*) and although the reference study (*Mishra et al., 2013*) described no defect for brain-silenced receptor, we found an impaired immediate preference on 1% but not on 2.5%.

We further proceeded to reassess sucrose detection of C2 by using an optogenetics approach as alternative to TNTE neuronal silencing. Optogenetic neuronal silencing of C2 using anion channel-rhodopsin (GtACR1) expression (*Mauss et al., 2017*) (see Materials and methods) led to an altered preference to sucrose 500 mM in larvae tested on 2.5% (*Figure 4D*).

All in all, our observations subscribe to the hypothesis that C2 and possibly other TOG neurons complement internal taste in sugar detection and that larvae can use distinct sensing paths—pharyngeal neurons being employed for sugar taste during food ingestion, whereas external taste neurons are participating in sugar detection especially when eating is not accessible (*Figure 5*). Thus far, our observations reveal partial roles in sucrose and fructose sensing associated to C2 or C5 neurons, and therefore it is probable that several TOG neurons might contribute to the output behavior in sugar response, either through a direct role or indirectly by computing a mixture of sucrose and other tastants as previously shown for C7.

## Discussion

In this study, we explore physiological and molecular characteristics of *Drosophila* larval taste neurons using a whole-organ imaging approach. We designed an experimental framework for calcium imaging in all GRNs comprised by the larval primary taste-sensing organ, the TOG (*Figure 1—figure supplement 1*, *Figure 1—video 1*). Our approach complements previous strategies based on targeting single or discrete subpopulations of neurons (*Croset et al., 2016*; *Sánchez-Alcañiz et al., 2018*; *van Giesen et al., 2016a*; *Kwon et al., 2011*; *Stewart et al., 2015*), allowing for an overview of activation patterns within the organ. As previously shown in *Drosophila* larvae, concurrent activation of two neurons can differ from their separate activation readout (*Hernandez-Nunez et al., 2015*) as specific sense neurons firing together might dictate the relevance of a stimulus and ultimately its behavioral output. Coherently, each tastant used here typically elicits response in several GRNs in support of a combinatorial sensing, while also raising the question about the redundancy in contribution of distinct neurons recruited in response to a given taste quality, to be elucidated by future studies.

For the tested substances at the concentrations chosen within this study, we observed roughly a 60–70% versus 30–40% division of unimodal versus multimodal responding GRNs (*Figure 2* and *Figure 2—figure supplement 1*), comparable with tuning proportions reported for mouse fungiform taste sensing cells (*Jisoo Han, 2018*). Taste stimulation with substance-modality groups showed a mixed tuning of responding neurons and, in particular, 30% of neurons stimulated with bitters and sugars were activated by both taste groups, in accord with the integration of opposed valence cues in single larval neurons. As previously shown for one GRN (*van Giesen et al., 2016a*), such types of sensory integration might allow the animal to evaluate mixed food sources at the level of the input neuron.

Interestingly, in animals stimulated with individual substances representative of different taste modalities (*Figure 2D*), up to 68% of responsive neurons were activated by only one tastant per animal. This could be due to the selected substance concentrations that could elicit weak responses undetectable at the chosen intensity threshold of response or at the imaging resolution used in this study. By varying the number and concentration of tested substances, we expect neuron tuning to be altered, as described on afferent taste neurons in mammals (*Wu et al., 2015*).

An interesting feature in larval taste physiology revealed by our observations was related to distinct response dynamics elicited by tastants in larval GRNs. Apart from canonical GCaMP fluorescence increase during stimulus, we also identified late neuronal activation that could be characterized as OFF-response with onset after stimulus application, but also GCaMP fluorescence drop during taste stimulus as perhaps deactivation response. Most notable deactivation signals were recorded in CDL2 neuron and most stereotypically to citric acid (*Figure 3* and *Figure 3—figure supplement 1B*). Interestingly, citric acid elicited concomitant activation in a neighboring cell named CDL1, indicating response patterns of activation-deactivation. The synchronous activation coupled with deactivation responses in adjacent neurons resemble the description of ephaptic signaling which implies that the electrical field of a responding cell generates hyperpolarization in the partner cell (*Jefferys, 1995*; *Faber and Pereda, 2018*). While ephaptic transmission has been described in the olfactory system

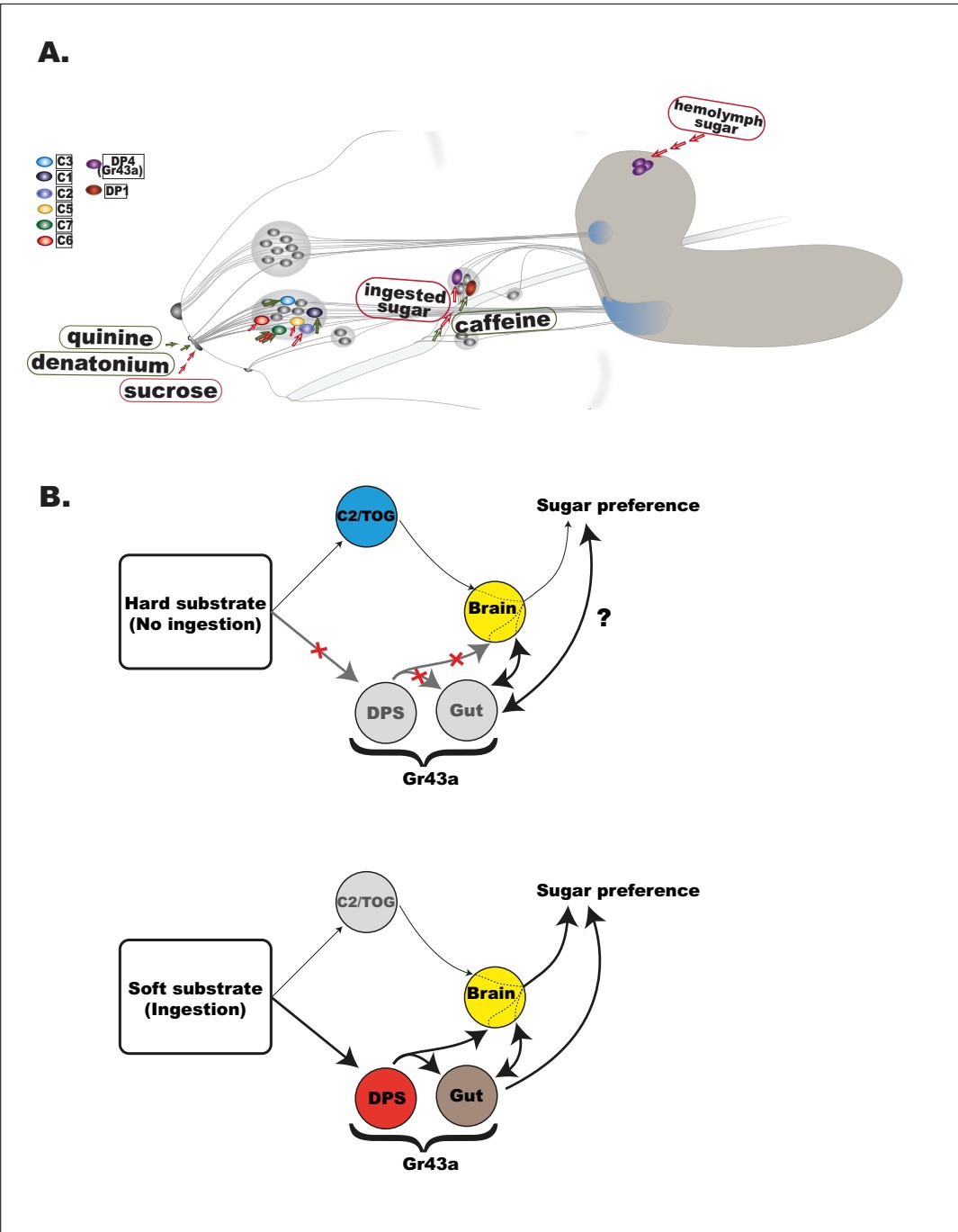

**Figure 5.** Peripheral taste sensing in *Drosophila* larvae—contribution to sugar and bitter taste of GRNs at different anatomical levels. (**A**) Schematic representation of the larval head and the chemosensory system. Bitter tastants detected at the level of external sense neurons are quinine in C3 neuron (*Apostolopoulou et al., 2014*) and denatonium in C7 and C1 (*van Giesen et al., 2016a*; *Choi et al., 2020*) but not caffeine, the latter being sensed through DP1 pharyngeal neuron (*Choi et al., 2016*; *Apostolopoulou et al., 2016*). Similarly, sugar taste seems to be also segregated at different levels of detection—brain *Gr43a* receptor neurons detect hemolymph sugar (*Mishra et al., 2013*), *Gr43a* expression in the pharyngeal organs (DP4 neuron) is central in sensing ingested sugar, while neurons like C2 and C5 contribute to sucrose preference at external level especially when food ingestion is less accessible (*Figure 4* and *Figure 4—figure supplement 1*). As the first gate for food evaluation, it is important to note the capacity of TOG neurons for integration of opposed valence taste wither within the same neuron as in C7 (*van Giesen et al., 2016a*), or by co-activation of a sugar sensing with a bitter sensing neuron as suggested for C2 and C1 (*Hernandez-Nunez et al., 2015*). (**B**) Schematic representation of the proposed model in larval sugar

*Figure 5 continued on next page*

*Figure 5 continued*

sensing. Upper panel: When unable to ingest as readily, larvae utilize C2 and/or other TOG neurons to guide sugar preference via a C2/TOG pathway. Gr43a is the main internal nutrient sensor, expressed in the pharyngeal taste neurons, proventriculus (gut), and brain. While the lack of ingestion prevents pharyngeal-expressing *Gr43a* neurons from contributing to immediate sugar preference, internal metabolic state signals may alter behavioral output. Lower panel: upon ingestion, internal sensing mechanisms take over sugar-associated preference most specifically for fructose, with a diminished role of external neurons (C2/TOG).

of the adult fly (*Su et al., 2012*; *Zhang et al., 2019*), it remains to be determined whether such lateral inhibition mechanism also occurs in *Drosophila* taste. Regardless of the underlying molecular mechanisms, deactivation events together with delayed and OFF-responses most probably enter in assembly of spatio-temporal codes as described for taste discrimination in the moth (*Reiter et al., 2015*), and remain an open fascinating question in the physiology of larval sensory neurons.

A large proportion of neurons responsive to one taste substance per animal were activated by sucrose and, interestingly, sucrose also elicited numerous responses in neurons activated by two or more tastants, in combinations such as sucrose+ citric acid, sucrose+ amino acid, sucrose+ high salt, and sucrose+ bitter (denatonium/quinine). It is possible that, at least in part, these neurons integrate sucrose response with other modalities for purposes of evaluating mixed dietary substrates for the purpose of avoiding dangerous food mixtures rather than having a direct role in sucrose sensing and output behavior, as previously indicated for C7 neuron (*van Giesen et al., 2016a*).

Nevertheless, since a direct participation of TOG neurons in sugar taste behavior has thus far not been shown, we here explored the roles of individual external gustatory cells. We started by testing C2, C6, and C7—neurons that have associated calcium responses to sucrose (*van Giesen et al., 2016a*). As expected, C7 has been linked to taste integration of opposed valence of sucrose and denatonium or quinine and we have not observed a defect in direct sucrose behavior in C7-silenced larvae (*Figure 4A*). Similarly, silencing C6 also did not result in behavioral defects. Known to have a role in $CO_2$ sensing (*Kwon et al., 2007*; *Jones et al., 2007*; *Faucher et al., 2006*), C6 is likely not involved in sucrose behavioral response per se, but prospectively important for integration of this taste in detecting fermenting substrates, and therefore would also have a role in computing mixed tastes, a hypothesis to be confirmed by future work.

We identified C2 to contribute in the attraction to sucrose. Interestingly, *Hernandez-Nunez et al., 2015* described a 'weak attractive response' in larvae when C2 was optogenetically activated alone but when co-activated together with C1 neuron, the C1-dependent repellent behavior was potentiated. This suggests that C2 neuron might be involved in sensing of an attractive cue while also enhancing the aversive behavior in association with a bitter neuron, as a possible mechanism for avoiding risky food mixtures. As silencing C2 does not result in a complete abolishment of sucrose attraction, additional TOG or external sense neurons might participate to sugar behavior. In support of this idea, we tested C5 neuron as a likely sugar-sensing candidate not expressing bitter-implicated receptors *Gr66a*/*Gr33a*, whose silencing also resulted in a trend for preference decrease compared to control larvae (*Figure 4—figure supplement 1A*).

The Gr43a receptor expressed in pharyngeal sense neurons and in the brain has a central role in sugar sensing in larval taste (*Mishra et al., 2013*). To fit in this context our observations on TOG calcium responses to sugar and a role of C2 neuron in sucrose sensing, we make distinction between external and pharyngeal taste. We compared larval behavior on easily ingestible 1% agarose concentration, as tested by Mishra et al., and on 2.5% substrate that is less accessible for eating (*Figure 4B*, *Figure 4—figure supplement 1B, B'*). Our results support the hypothesis that C2 neuron in the TOG senses sucrose and contributes to behavior prior to and in absence of eating, whereas Gr43a is the main sugar receptor but requires food ingestion. The overrun of *Gr43a* guided behavior during food ingestion may be due to the lesser need for external sugar sensing when it has already been located and ingested, thus switching to an alternative internally guided pathway for sugar preference behavior (*Figure 5B*). Also, the reminiscent behavioral defects of *Gr43a* mutant to fructose sensing on 2.5% substrate (impaired ingestion) could be due to *Gr43a* expression in several internal tissues (pharynx, proventriculus, and brain) and to its role as hemolymph fructose sensor. It has been shown that hemolymph carbohydrates influence larval feeding behavior through insulin signaling and

SLC5A11 (*Ugrankar et al., 2018*), but it remains to be determined if and how does a *Gr43a* mutant correlate with metabolic pathways and possibly interfere with sugar preference output behavior.

In the adult, *Gr43a* acts as the main fructose hemolymph sensor but also associates a role in feeding suppression in satiated animals (*Miyamoto et al., 2012*). Other moderators, such as the *Ir60b* neurons, carry a role in limiting sugar consumption (*Joseph et al., 2017*). Our study adds a new piece of evidence to deconvolute sugar sensing mechanisms in the larva while setting down a path for future investigation of sugar detection at this developmental stage.

The observations reported here in relation to taste sensing and cellular segregation of behavior can be paired with previous data for a depiction of sugar/bitter taste detection at the periphery (*Mishra et al., 2013*; *Apostolopoulou et al., 2014*; *Kim et al., 2016*; *van Giesen et al., 2016a*; *Choi et al., 2016*; *Choi et al., 2020*; *Figure 5*): in the TOG larvae detect quinine in C3 neuron, denatonium in C1, sucrose in C2 and C5, but also mix of sucrose and denatonium/quinine in C7 and also possibly a mix with sucrose in $CO_2$-detector C6; at the level of pharyngeal neurons dietary sugar is sensed in DP4 (expressing *Gr43a*) and caffeine in DP1; in the brain, hemolymph sugar is detected by *Gr43a* receptor neurons. Based on this summary, taste sensing in *Drosophila* larva would entail cell multimodality for food mixture processing, combinatorial taste through co-activation of neurons of distinct valence, and anatomical or molecular segregation of specific tastant detection. Evidently, despite its numerical simplicity, the study of taste in the fruit fly larva reveals numerous layers of complexity.

The present study offers a sample overview for physiological taste responses in the peripheral taste organ of the *Drosophila* larva. Our data illustrates the sensing capacity and complexity of responses at this level as revealed by multimodality of response in a fraction of neurons and by distinct temporal dynamics in activation. We also determine for the first time roles in sugar sensing behavior of larval external taste neurons, by segregating external and pharyngeal taste. We thus identify C2 as contributor to sucrose taste and open directions for further investigations on additional external sugar sense neurons and the query of their redundancy, role in computing substance mixtures, as well as specific molecular mechanisms and receptors involved in sugar detection.

## Materials and methods

### Fly stocks

We used the following fly strains: *nSyb*-Gal4 (gift from Shanhaz Lone), UAS-*GCaMP6m* (Bloomington #42748), UAS-*H2B:RFP* (gift from Boris Egger), GMR57BO4-Gal4 (Bl. #46355), *Gr94a*-gal4 (Bl. #57686), *Gr21a*-Gal4 (Bl. #23890), *Gr22e*-Gal4 (Bl. #57608), *Gr66a*-Gal4 (Bl. #57670), *Gr33a*-Gal4 (Bl. #57623), *Gr59d*-Gal4 (Bl.57653), *Gr32a*-Gal4 (Bl. #57622), *Gr97a*-Gal4 (Bl. #57687), *Gr59e*-Gal4 (Bl. #57655), UAS-*myr:GFP* (Bl. #32198), and UAS-*TNTE* (Bl. #28837). Gr43a lines: *Gr43a*-KI-Gal4 mutant, *Gr43a*-KI-GAL4; UAS-*Gr43a* and *Gr43a*-KI-GAL4; UAS-*Cha7.4kb* were kindly offered by Hubert Amrein.

### Chemicals

Chemicals of highest purity available were chosen for chemicals: L-Alanine (Sigma-Aldrich, 56-41-7), L-Glutamine (Sigma-Aldrich, 56-85-9), L-Aspartic acid (Sigma-Aldrich, 56-84-8), L-Glutamic acid (Sigma-Aldrich, 56-86-0), L-Asparagine (Sigma-Aldrich, 70-47-3), L-Proline (Sigma-Aldrich, 147-85-3), L-Tyrosine (Sigma-Aldrich, 60-17-4), L-Threonine (Sigma-Aldrich, 72-19-5), L-Phenylalanine (Sigma-Aldrich, 63-91-2), L-Lysine (Sigma-Aldrich, 56-87-1), L-Arginine (Sigma-Aldrich, 74-79-3), L-Histidine (Sigma-Aldrich, 71-00-1), L-Serine (Sigma-Aldrich, 56-45-1), L-Valine (Sigma-Aldrich, 72-18-4), L-Cysteine (Sigma-Aldrich, 52-90-4), L-Methionine (Sigma-Aldrich, 63-68-3), L-Leucine (Sigma-Aldrich, 61-90-5), L-Isoleucine (Sigma-Aldrich, 73-32-5), L-Tryptophan (Sigma-Aldrich, 73-22-3), L-Glycine (Roth, 3187.3), D-Sucrose (Sigma-Aldrich, 57-50-1), D-Glucose (Sigma-Aldrich, 50-99-7), G-Fructose (Fluka, 57-48-7), D-Trehalose dihydrate (Roth, 9286.1), D-Arabinose (Sigma-Aldrich, 10323-20-3), D-Maltose monohydrate (Sigma-Aldrich, 6363-53-7), D-Mannose (Sigma-Aldrich, 3458-28-4), D-Galactose (Roth, 4979.1), D-Cellobiose (Roth, 6840.3), Lactose monohydrate (Roth, 8921.1), Quinine hemisulfate salt monohydrate (Sigma-Aldrich, 6119-70-6), Denatonium benzoate (Sigma-Aldrich, 3734-33-6), Caffeine anhydrous (Fluka, 58-08-2), Coumarin (Sigma-Aldrich, 91-64-5), Sucrose octaacetate (Sigma-Aldrich, 126-17-7), Lobeline hydrochloride (Sigma-Aldrich, 134-63-4), Theophyline (Sigma-Aldrich, 58-55-9), Strychnine (Roth, 4843.1), Potassium chloride (Merck, 7447-40-7), Sodium chloride (Sigma-Aldrich,

7647-14-5), Citric acid (Sigma-Aldrich, 77-92-9), Agarose standard (Roth, 3810.4), and Brilliant Blue FCF (Wako, 42090).

## Immunostainings

Third instar larvae (4 days after egg laying) were washed and dissected in ice-cold PBS. Tissue containing chemosensory neurons was fixed in 3.7% formaldehyde for 20 min at room temperature. After fixation, incubation in primary antibodies was performed at 4°C overnight, followed by washing steps and incubation in secondary antibodies at 4°C overnight, and finally, in Vectashield antifade medium (Vector Laboratories) for at least 1 hr before mounting the samples on microscope slides. After each step described above, PBST (PBS 0.3% Triton X-100) was employed for three consecutive plus three 30-min washes at room temperature. Primary antibodies: chicken anti-GFP (1:1000, Abcam, ab13970), rabbit anti-DsRed (1:1000, Clontech, No. 632496), and rat anti-ELAV (1:30, DSHB, No. 7E8A10). Secondary antibodies conjugated with Alexa Fluor fluorescent proteins (488, 568, and 647) were used in dilution of 1:200 (Molecular Probes nos. A-11008, A-11039, A-21244, A-21247, and A-11011).

Confocal images were acquired on a Leica TCS SPE-5 confocal, using the 40× oil immersion objective, at 0.8–1 µm slice thickness. Images were assembled using Fiji (*Schindelin et al., 2012*), Adobe Illustrator, and Adobe Photoshop.

## Calcium imaging

The semi-intact sample for calcium-imaging recordings was prepared as previously described (*van Giesen et al., 2016b*). Third instar larval heads comprising chemosensory neurons, the brain and the connecting nerves were dissected in AHL (adult hemolymph-like) saline solution. After mounting into the microfluidic chip, the sample was connected to the tubing and the micropump setup. For whole-organ TOG recordings, UAS-*GCaMP6m* (*Chen et al., 2013*) was driven in all neurons and *RFP* was expressed in nuclei using UAS-*H2B:RFP* (*Egger et al., 2007*).

## Microscopy settings

Calcium imaging was performed using a Visitron VisiScope CSU-W1 and Nikon Ti-E inverted spinning disk confocal microscope. The images were captured using a Photometrics Evolve 512 Electron Multiplying Charge-Coupled Device (EM-CCD) of 16×16 µm² pixel size, pixel binning 2×. The total field of view (FOV) diameter was 55 µm. Laser power for both channels was set at 150 mW, and the exposure time for each Z-section was set at 80 ms. Imaging parameters: 25 slices/stack; 1.5 µm each slice for a total depth size of 37.5 µm; time acquisition speed 0.5 time points per second (7.5 fr/s) or 2 seconds per stack.

For each recording, an RFP stack (one time point) was acquired separately from the GCaMP time series consisting in 60 time points per taste stimulation. Recordings were processed in Fiji/ImageJ (*Figure 1—figure supplement 1*) and transferred to Imaris 9.6.0 software for visualization and cell segmentation (*Figure 1—video 1*).

## Taste stimulation

Each tastant was administered using a micropump (mp6, Bartels Mikrotechnik) controlled in VisiView software through built-in macros. For each taste stimulation, macros were designed for 2-min length recording: 60-s pre-wash followed by 30-s stimulation and a final 30-s wash. Millipore water was used as solvent for taste solutions and as washing control.

### Taste stimulation with groups of tastants

We compiled two groups of sugars, each containing five sugars at a concentration of 100 mM each: monosaccharides (fructose, glucose, arabinose, mannose, and galactose), and disaccharides (sucrose, trehalose, maltose, lactose, and cellobiose). Amino acids were prepared as previously described by *Park and Carlson, 2018* (*Table 1*) in four groups each at 10 mM concentration: group A (valine, leucine, isoleucine, methionine, tryptophan, and cysteine), group B (alanine, phenylalanine, glycine, and proline), group C (arginine, lysine, aspartic acid glutamic acid, and histidine), group D (serine, threonine, asparagine, and glutamine), and tyrosine was included in group B at 1 mM concentration,

limited by its solubility. Salt taste groups contained NaCl and KCl in equal proportions at a total concentration of 50 mM (low salt) and respectively 1 M (high salt). Bitter tastants were randomized in two groups of four substances each, with a final concentration of 22 mM and respectively 13 mM, restricted by the solubility of certain substances such as lobeline or strychnine reduced to 1 mM instead of 10 mM (*Table 1*): DSoTC group (denatonium benzoate, sucrose octaacetate, theophylline, and coumarin), and respectively QLSC group (quinine, lobeline, strychnine, and caffeine). The order of chemicals was randomized. The response threshold was chosen at 20% and the number of responding neurons within each trial of stimulation is specified in sheet 1 of the *Figure 2—source data 1*.

## Taste stimulation with series of individual tastants

We used two series of tastants (*Table 1*), each containing five substances from five different taste categories: sucrose 500 mM for sweet taste, denatonium benzoate 10 mM (series 1), or quinine hemisulfate 5 mM (series 2) for bitter taste, NaCl at 1 M for high salt (series 1) or at 100 mM for low salt (series 2), valine 100 mM (series 1), or arginine 100 mM (series 2) as amino acid and citric acid for sour taste. Each recorded animal was stimulated with one tastant from each taste category and a total of 15 TOG organs were recorded, 7 within the first series of stimulation and eight within the second series. The order of chemicals was randomized. Neurons responding above the amplitude threshold of 20% are listed in *Figure 2—source data 2*.

## Data processing and analysis of whole-organ physiological recordings

RFP stacks and GCaMP recordings were treated for deconvolution using the third-party program Huygens (Scientific Volume Imaging). A Fiji/ImageJ plugin was written to duplicate the RFP signal onto all time points of the GCaMP recording (*Figure 1—figure supplement 1*; *Figure 1—figure supplement 1—source data 1*). For correction of drift or animal movement, a 3D drift correction plugin (*Parslow et al., 2014*) was run on the GFP signal before merging the two channels. This software code allows for decent amendment of z-drift of hyper-stacks. Subsequently, the drift-corrected GFP and the original duplicated RFP stack were merged and the misalignment between the two signals was adjusted using an in-house Fiji/ImageJ macro for manually guided realignment on the x-y axis (*Figure 1—figure supplement 1*; *Figure 1—figure supplement 1—source data 1*). A final 3D drift correction was performed on the aligned two-channel hyperstack for better stabilization. If at this step the drift or animal movement rectification results were not satisfying, the recording was discarded. The resulting recording was transferred in Imaris 9.6.0 for automatic segmentation with spots of 3.5 μm diameter (*Figure 2A*, *Figure 1—video 1*). Each uniquely identified neuron through spot detection received a random identity number and a corresponding fluorescence trace over time (*Figure 1—figure supplement 1*).

Downstream analysis was performed with R (*R Development Core Team, 2017*; *Wickham et al., 2018*). Normalization of fluorescence values followed the formula $(F_t−F0)/F0$, where $F_t$ is the intensity at time point t and F0 the baseline calculated as average intensity of 10 frames prior to stimulation. Response amplitude was calculated as the difference between the peak intensity after stimulation (average of 5 frames near the maximal fluorescence value) and F0. All traces were manually validated to exclude false positive responses. Any of the following criteria was sufficient to exclude traces as possibly false positives: the baseline signal was unstable or the baseline variance was higher than the calculated response amplitude; the fluorescence rise upon stimulation could have been caused by an artifact; a delocalized fluorescence signal could have been captured by the cell spot, compromising the measured response. For further analysis, a response threshold was set at 20% DF/F0. To calculate taste integration percentages (*Figure 2*), for each data set, we pooled together all neurons eliciting at least one response above 20% in amplitude and determined the fraction activated by one or by multiple tastants/taste groups per animal.

## Cellular mapping

This step was performed entirely manually for recordings with individual tastant stimulation (*Figure 2D and D'*). Responsive neurons were compared to their neighbors and to the reference four neurons localized anterior-laterally named ALN group (Anterior-lateral-neurons, *Figure 3B*). For each recorded response above the 20% threshold, the activated cell was thus compared with the neighboring cells for determining an approximate location within the organ and then compared across animals with

responding neurons with similar approximated position. If one of these comparison steps was incon-clusive, the cell response was not considered for the mapping. In *Figure 3C*, we plotted responses that were each manually mapped in at least four separate recordings (exception for responses to bitter substances where plotting was considered for at least three separate recordings).

Figures were generated in RStudio (*R Core Team, 2015*) using 'ggplot2,' 'reshape2', and 'scales'' packages (*Wickham, 2007*; *Wickham, 2009*; *Wickham, 2019*) or in GraphPad Prism 7 and finally adjusted in Adobe Illustrator and Adobe Photoshop.

## Larval behavior

L3 4-day-old larvae grown on stable conditions of 25°C and 12-hr light/dark cycle, were picked from the food (feeding larval stage) with a brush, washed in tap water, and tested for taste preference in a behavior dark room to exclude additional stimulation of the animal.

## Two-choice assay

Petri dishes (94 mm) were prepared on the day of testing as follows: plain agarose at 2.5% concen-tration was poured in the dish; after hardening, a middle straight line was used to cut the substrate in two halves; one half was discarded and replaced with agarose of the same concentration containing a test substance. Agarose plates were left to reach ambient temperature for at least 1 hr before testing. For comparison with previously reported results for Gr43a sugar sensing role, agarose plates were prepared identically at 1% concentration. For each test 30 larvae: after washing, larvae were placed on a middle line region of the plate and left to wander. After 5 min, animals were counted on each side of the plate and the preference index was calculated as follows:

$$PI = (n° \text{ larvae on test substance} - n° \text{ larvae on plain agarose})/n° \text{ total larvae}$$

For experiments consisting in several time points, counting was repeated at 2 min, 5 min, and 15 min. At the time of counting, animals that were found on the middle region defined as a 1-cm width-line between the two halves were not considered on either choice side and therefore taken as neutral. For example, if among 30 larvae, 19 were found on the sucrose side, 7 on the empty half, and 4 on the middle stripe region, the index is calculated as PI=(19−7)/30=0.4.

For each experiment, tests were repeated for N≥7 (typically N=10—see *Figure 4*, *Figure 4—figure supplement 1* and respective source data files for details). GraphPad Prism 7 was used for data visu-alization and for statistics. Mann-Whitney test was applied to statistically compare the test genotype with each control line (p-value<0.05; **p<0.01; ***p<0.001).

## Optogenetics

The optogenetics experiments were performed as described in *Mauss et al., 2017*. Briefly, larvae were assayed in petri dishes half coated with 2.5% agarose and half with 2.5% agarose containing 500 mM sucrose. Tubes containing larvae were exposed to 527 nM color light emitted from a projector (Epson LCD Projector model H763B, default settings) situated 70 cm above the experimental space for 2 hr prior to experimentation for stimulated experiments, or conversely kept in darkness for 2 hr for unstimulated (control) tests (*Figure 4D*). The number of larvae on each substrate was counted for the preference index calculation at 2, 5, and 15 min after placement.

## Food ingestion test

Petri dishes were filled with 1% or 2.5% agarose mixed with sucrose 500 mM and Brilliant Blue dye 2%. Larvae were left to wander on the plate for 2, 5, 15, or 30 min, before being removed and washed in tap water. The blue dye present in their abdomen tracks the agarose larvae were able to ingest on the two different concentrations.

## Acknowledgements

The authors thank the Sprecher lab for constructive discussions, and especially to Cornelia Fritsch for lab management support. The authors are grateful to Boris Egger for guidance on microscopy and data analysis. Many thanks for fly stocks to Hubert Amrein, Shanhaz Lone, Bloomington *Drosophila* Stock Center.

## Additional information

### Funding

| Funder | Grant reference number | Author |
|---|---|---|
| Schweizerischer Nationalfonds zur Förderung der Wissenschaftlichen Forschung | 310030_188471 | Simon G Sprecher |
| Novartis Stiftung für Medizinisch-Biologische Forschung | 18A017 | Simon G Sprecher |

The funders had no role in study design, data collection and interpretation, or the decision to submit the work for publication.

### Author contributions

G Larisa Maier, Conceptualization, Data curation, Methodology, Visualization, Writing – original draft, Writing – review and editing; Nikita Komarov, Formal analysis, Investigation, Methodology, Writing – review and editing; Felix Meyenhofer, Software; Jae Young Kwon, Conceptualization, Supervision; Simon G Sprecher, Conceptualization, Funding acquisition, Supervision, Writing – original draft, Writing – review and editing

### Author ORCIDs

G Larisa Maier ![ORCID] http://orcid.org/0000-0002-4486-8296
Simon G Sprecher ![ORCID] http://orcid.org/0000-0001-9060-3750

### Decision letter and Author response

Decision letter https://doi.org/10.7554/eLife.67844.sa1
Author response https://doi.org/10.7554/eLife.67844.sa2

## Additional files

### Supplementary files

• Transparent reporting form

### Data availability

All data is available as part of the submitted manuscript.

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
