## [Editor Report]

This paper provides an alternative theory about taste coding in larval insects. The authors use a volumetric imaging method to capture neural activity from an entire taste sensory organ in *Drosophila* larvae. Their results suggest that taste coding in *Drosophila* larvae might be different than adult flies, and use multimodal neurons and neural dynamics to represent taste information in the larval brain. Understanding taste coding in *Drosophila* larvae might provide further understanding to how food is encoded in the larval insect brain.

---

## [Decision Letter]

**Decision letter after peer review:**

Thank you for submitting your article "Taste sensing and sugar detection mechanisms in *Drosophila* larval primary taste center" for consideration by *eLife*. Your article has been reviewed by 3 peer reviewers, including Sonia Sen as Reviewing Editor and Reviewer #1, and the evaluation has been overseen by K VijayRaghavan as the Senior Editor.

Essential revisions:

The methods used in this manuscript are useful and the findings extremely interesting. There were a few points that concerned us, and we list here experiments that will help support the claims made in manuscript better.

– Functional Imaging

1. We would like to get a sense of the variability in responses to the tastants – both inter- and intra- individual variability. To illustrate this, we are asking the authors to (1) represent the data as *per neuron, per animal*, and (2) show more traces (as in 2D) for individual cells from individual animals across trials with different tastant stimulations.

2. To irrefutably assign taste response profiles to each of the neurons, we're also suggesting (1) dose responses to each tastant, and (2) swapping the order of tastant presentation while recording responses.

3. Central to better describing inter-individual variability will be the unambiguous identification of individual GRNs across individuals. This needs to be shown more definitively and we recommend a uniformed naming system to avoid confusion. One possible way they could do this is by labelling known GRNs in the whole organ imaging. If only a small number of GRNs can be so identified, perhaps the authors could focus on just these for their analyses.

4. The methods and criteria used for image acquisition and processing needs to be described better.

– Behaviour

The authors' central thesis – peripheral sweet neurons>hard food; internal sweet neurons>soft food – hinges on the result that on hard food, despite the absence of 43a from the pharyngeal neurons, larvae still show a preference for sugar. Presumably larvae are not eating during the 2-5 minute window (4B) and so the brain 43a receptors cannot driving this preference for sugar. They reason, it must be peripheral neurons that are responsible for this. We have three issues with this central finding and their interpretations:

1. *Sugar or Texture?* There's a possibility that mechanosensation (response to texture of the food) might be playing a role in this observation. The authors could demonstrate the specificity of this response by performing a dose response curve with sucrose (we recommend sticking to sucrose as it shows more robust PIs). So, we recommend that they perform the experimental conditions in 4B(bottom) with sucrose at different concentrations.

2. *The 43a null*: If the peripheral neurons were driving this behaviour, the 43a null mutants should show at least a residual sugar preference, which they don't on either soft or hard food. It's possible that the authors are missing this in the low PIs of 4B. The above experiment should uncover this.

3. *Peripheral neurons on soft food only?* What keeps the peripheral neurons from driving sugar preference on soft food in the absence of 43a? If the authors are suggesting that the peripheral sugar neurons are gated by both *sugar and texture*, they would need to demonstrate this through functional imaging. And in that case, activating the peripheral-GRNs in the brain-only 43a animal should induce sugar feeding on soft food.

In the individual reviews below, the authors will find more specific points that pertain to the broad points mentioned here. Could the authors please pay attention to them as they address these points?

*Reviewer #1:*

The authors use a very nice microfluidic set-up to monitor neuronal activity in an ex-vivo larval TOG while simultaneously bathing it in a variety of tastants. They show a surprisingly high level of multimodality in the GRNs – >30% respond to 2-4 tastants. On occasion, these overlaps were of opposing hedonic value and so the authors propose a high level of signal integration in the larval taste organ. They seem to suggest that this could occur via temporally distinct responses in the GRNs.

Subsequently, the authors focus on sugar sensing in this peripheral organ. Interestingly in larvae, the 43a receptor that senses sugars is not expressed in the periphery but in the pharynx (where it can detect sugar in the ingested food) and in the brain (where it can detect haemolymph sugars). Here the authors demonstrate that many of the peripheral GRNs can respond to sugars. They silence 3 GRN classes to find that one of them is necessary/involved in sugar sensing.

They next turn to addressing the issue of peripheral vs internal sensing of sugar. They perform behavioural experiments to calculate preference for sugars while silencing the peripheral GRN or mutationally inactivating GR43a. They perform these experiments in soft and hard food and suggest that in soft food (when the animal can easily ingest) the internal sugar sensors are utilised. On hard food however, they suggest that the internal sensors are not as important, but that the peripheral GRNs involved in sugar sensing are necessary. This is an interesting idea.

My primary concern with this manuscript is the section dealing with the internal vs peripheral sensing of sugars. It's possible that some of my concerns arise out of the manner in which these results are reported and discussed, nevertheless, I will list them out here:

– The preference indices in figure 4B seem very low: 0.2. At such low PIs I wonder if the authors might mistakenly interpret a lower π for a 'no-preference'. In general I did not understand the reasons for using the various sugars at differing concentrations in this section.

– The whole animal 43a mutant on both hard and soft food shows a complete inability to detect sugar. If this is true, then it suggests that the pheripheral neurons are NOT involved in sugar sensing. Do the authors agree with this interpretation?

– The brain-only 43a mutant data makes the whole animal mutant data more difficult to interpret. If the difference between soft and hard food is a difference in time taken to ingest the food (as shown in Figure 4B), then there should be no difference in the trends of sugar responsiveness except in the axis of time. I have not understood why the authors think that on soft food the peripheral GRNs don't compensate for the lack of pharynx-43a to drive sugar feeding. Can the authors please comment on this?

*Reviewer #2:*

Maier et al., investigates neural basis of sugar perception/coding in *Drosophila* larvae using in vivo calcium imaging and behavior. Authors developed a volumetric imaging method to capture taste responses in all of the TOG neurons in larvae. The whole organ imaging method developed here will be a very useful technique for *Drosophila* larval chemosensory imaging. However, the imaging method and data analysis pipeline to identify cellular identity is not clearly explained in the manuscript. It is important to provide a metric for how effective and reliable cellular identity classifications are and the variations among individual larvae. The volumetric imaging results indicate that TOG neurons can respond to one modality or multiple modalities. The neural response dynamics are also variable. Some neurons show ON responses while some show OFF-responses or inhibition upon tastant stimulation. Furthermore, the authors test the function of TOG neurons in regulating sugar preference and found that the C2- TOG neuron is important for discriminating sugar when food is more viscous and harder to ingest. Overall, I think the volumetric imaging method is promising however, the imaging method and data analysis pipeline is not clearly explained and no metric has been provided to show how effective 3D cellular identity analysis is. The behavioral experiments regarding C2 neurons are not consistent with the previously published functional imaging analysis from the same group. Further functional imaging experiments are necessary to support the claims in the paper.

1. The whole organ imaging method developed here will be a very useful technique for *Drosophila* larval chemosensory imaging. However, the details of image capture and processing is somewhat vaguely explained in the manuscript. Multiple details and metrics are missing about how imaging is done. What is the laser power used in these experiments? What is the frame rate for each volume scanned? What is the size of each volume, field of view etc. The movement and the drift correction for calcium imaging makes sense, however more detailed explanation about how they process imaging data is necessary. For example, in Figure Supplement 1, the authors are correcting GFP channels using the position information from the RFP channel and their method is able to handle the xy-plane. However, it is not clear how they handle z-drift. Did the post-processing also apply for the movement and drift correction in the Z-axis? What is the error rate for x-y or z correction? It would be good to know how accurate their methods are in identifying the same cell identities across animals. The authors need to expand the methods section and explain clearly how volumetric imaging and data processing was done in the paper. This is key for accurate interpretation of cell identification and the calcium imaging results.

2. Is figure 2B and 2C showing the percent responsive cells recorded across multiple animals or the percent responsive cells within an animal? I think it is important to clarify this point. What is the variance between individuals? I would like to understand how reliable cell identification and functional imaging analysis is across animals. This can be achieved if authors show data per animal rather than lumping all the data across the animals together. Another reason to visualize individual animal responses is to investigate whether the order of tastant stimulation is a critical factor for larval taste perception: For a neuron that can respond to bitter and sugar, would the sequence of stimulation matter? Will the neuron change activity if it was stimulated first with a bitter compound then a sugar compound? What about the opposite? I strongly encourage authors to show more traces for individual animals across trials with different tastant stimulations as done in Figure 2D. This will help the reader understand what is the variation in taste responses across animals and it will also increase confidence for the consistency of authors' volumetric imaging method.

3. In figure 3A, the authors show different calcium dynamics upon tastant stimulation in different TOG neurons. They have observed, off responses and inhibition of neural activity in certain neurons upon stimulation. I would like to see what is the percentage of OFF-responses and inhibition observed in these experiments. Are these events frequent or rare? Are they seen in all animals or some of the animals used in the calcium recordings? It is important to understand whether these are biologically relevant neural responses or artifacts seen in some of the animals used in the experiments.

4. In Figure 3, authors replot the results from Figure2 this time by identifying cellular identity using anatomical position. This is something similar to what I was trying to get to in my previous comment (see comment 2). But even in plots shown in Figure 3C, it is hard to understand how successful authors' methods are in classifying each neuron type in each animal. For example, how many larvae in total have been tested in Figure 3C and in what percentage of the animals they were able to identify each neuron type and what is the variation in neural responses among each animal. I think this is a very critical point that needs to be clarified in the paper variation of taste responses among animals and a metric to show how successful cellular identification is across all the animals tested. If my understanding is correct, the total number of tested organs/animals is 15, however the data plotted in Figure 3C does not correspond to this number. Please explain what is the success rate to define a cell type across animals.

5. The authors mention that the only known sugar receptor gene Gr43a is not expressed in TOG but TOG neurons can respond to sugars. This is a puzzling finding. Is the expression analysis of Gr43a receptor based on transgenic reporters or single cell sequencing? In other words, is there other evidence that TOG does not express any known sugar receptors in larvae?

6. In figure 4, authors performed behavioral experiments to test the function TOG sugar sensing in larvae. Inhibiting one of the TOG neurons, C2 causes defects in sucrose preference when agarose ss 2.5 % but not 1%. They claim the external sugar sensing via TOG sensory neurons is critical when food is not soft enough to ingest. These are interesting hypothesis however, in their previous study, authors have shown C2 neurons respond weakly to 500mM sucrose but not to any other sugar. However, in this manuscript the stronger behavior phenotype is seen in 250mM fructose preference upon silencing of C2 neurons. It is not clearly to me what is the mechanism here. Please explain.

7. In general, I think it will help to use one nomenclature for the paper or provide a table matching C-X identity to cellular identities in Figure 3C and list which compound(s) each neuron responds to and indicate the type of response; ON, OFF or inhibition.

8. The manipulations for larval GRNs in the manuscript are done chronically by expressing TNT-E. This might impact the development and wiring of the larval chemosensory system and cause problems in sugar detection unrelated to GRN function. I think using a conditional inhibitor of neural activity such as shits or newly developed ontogenetic inhibitors, GtACRs, might be better to use here to show conditional inactivity of the neurons result in similar defects in sugar sensing.

*Reviewer #3:Drosophila* larvae have been put forth as a good model to study taste because of the simplicity of the system (in terms of the number of organs and neurons) and efficient behavior assays. However, an understanding of taste coding in the larval system lags behind that described for the adult stage, in which most taste sensilla/neurons are accessible for recordings or imaging. Maier et al., seek to address this gap by developing a whole organ calcium imaging approach to assess responses of neurons in the larval terminal organ to sweet, bitter, salt, amino acid, and acid tastants. From imaging 15 organs, the authors describe responses to a set of sweet, bitter, salt, amino acid, and acid tastants, collectively represented in neurons that appear to be tuned to one, two or multiple categories of tastants.Although the new approach has promise, the current study makes a limited conceptual advance and the number of positive responses used to ascribe responses to neurons is small. Overlap of tastant sensitivities has now been described, and more rigorously so, in the adult stage and in the authors' own previous work focusing on genetically labeled subsets of taste neurons. The identification of responsiveness to sucrose, which may be somewhat unexpected given the absence of sweet Gustatory receptors (Grs) in larvae, has also been previously described for a few neurons, one of which is further studied in behavior assays.

The imaging analysis is followed by investigation of the potential roles of sugar sensing neurons in the periphery. While this analysis provides further support for a previously identified peripheral neuron (C2), it does not convincingly parse contributions of internal and external neurons.

Specific weaknesses are elaborated via points below and could be addressed either with additional experiments or textual modifications.

Calcium imaging:

– Neuron categories were determined by imaging responses to only a single concentration of each tastant; more careful dose response analysis may alter the conclusions. This possibility has not been considered.

– There is no effort to reconcile the imaging results with predictions from previous studies. For example, the number of bitter-responsive neurons is 1-2 per larva. However, bitter-GAL4 lines have been mapped to many more neurons in the TO.

– The correspondence between neurons named from previous GAL4 mapping experiments to those named by position in this study is not at all clear. A better comparison between the two could be presented.

–The criteria that were used to manually exclude false positive responses in calcium imaging experiments (Line 475-476) have not been described.

– A response was ascribed to an identified neuron if the position was the same in >3 organs. This threshold seems low, considering that 7-8 organs were tested with each stimulus. In the table for Figure 3, for instance, several tastants have n=3. Is this from a higher number of tastant applications? If so, what was observed in the other cases – that a response could not be ascribed to the identified neuron or that there was no response at all?

Behavior:

– There are no indications of statistically significant differences between the genotypes in Figure 4B; the legend only mentions that the preference of Gr43a mutants is not significantly different from zero. Finally, data in Figure 4C are compared with Mann-Whitney tests, but ANOVA should be used instead.

– Whether sucrose response of internal neurons is affected in Gr43a mutants has not been tested (relevant to Figure 4C).

– In Figure 4B right, assuming that Gr43a mutant controls are not significantly different from zero, whereas Gr43a "brain only" are, I don't know how to interpret a potential role for Gr43a brain neurons in preference for 2.5% agarose food, at 2 minutes when the larvae are not consuming anything.

– The calcium imaging analysis is at the cellular level, not looking into contributions of receptors. Manipulating cells rather than receptors for behavior experiments would allow for better comparison between the two aims of the study. If the Gr43a mutant is the only way to go, suggested experiments are:

1. Test if the complementary brain ablated genotype (Gr43a internal taste only) yields the expected phenotypes on 1% and 2.5% agarose.

2. Test if peripheral sugar responses are affected in Gr43a mutants.

---

## [Author Response]

Essential revisions:The methods used in this manuscript are useful and the findings extremely interesting. There were a few points that concerned us, and we list here experiments that will help support the claims made in manuscript better.– Functional Imaging1. We would like to get a sense of the variability in responses to the tastants – both inter- and intra- individual variability. To illustrate this, we are asking the authors to (1) represent the data as per neuron, per animal, and (2) show more traces (as in 2D) for individual cells from individual animals across trials with different tastant stimulations.

Thank you for your observation, we indeed agree that showing more traces helps in better grasping the data and the potential entailed by imaging tools described here.

We have therefore added more traces in Figure 2D per neuron to different tastants within the same organ in 2 separate representations. Also, in Figure 3 we are including traces per mapped neuron to a given tastant from three different organs. The previous data representation of Figure 3 was moved to Figure 3-figure supplement 1 with a more stringent threshold for plotted tastant responses per cell. All responses recorded as above 20% in amplitude included in the paper are available in the supporting information (source data excel tables).

2. To irrefutably assign taste response profiles to each of the neurons, we're also suggesting (1) dose responses to each tastant, and (2) swapping the order of tastant presentation while recording responses.

1. This is an excellent suggestion, as proved by the well documented example in *Drosophila* of low vs. high salt concentrations that activate distinct taste cell populations for divergent output behaviors. Nevertheless, manually mapped neurons in response to a given tastant across separate organs are displayed in Figure 3 and Figure 3-Supplement1 as proof of concept and we believe that more ambitious attempts for neuronal mapping and to a range of tastant concentrations should be assisted by tissue registration techniques. We propose that irrefutably assigning response profiles to each neuron does not make the main object of this study, but we present a valuable whole organ analysis tool in assisting future efforts using complementary necessary techniques for neuron labeling towards this purpose. Instead, our data aims at providing a grasp of the variety of recorded responses to 5 tastants from distinct modalities in average in the larval external taste organ.

We hope that reviewers accept our reasoning, as a dose response analysis on a manually mapping approach of cell identities would be extremely challenging.

2. The tastant order swapping is an important point and this is included in the protocol design of whole organ recordings, as the order of tastant presentation was randomized for each organ. Also, we have now additionally performed single neuron calcium imaging in C7 – neuron known to respond to both sugar and bitter tastants (van Giesen et al., 2016a) – to exemplify the influence of order presentation on sucrose-elicited responses. We have found no significant difference between sucrose response before or after stimulation with denatonium in this neuron, as shown in Figure 2-Supplement 2.

3. Central to better describing inter-individual variability will be the unambiguous identification of individual GRNs across individuals. This needs to be shown more definitively and we recommend a uniformed naming system to avoid confusion. One possible way they could do this is by labelling known GRNs in the whole organ imaging. If only a small number of GRNs can be so identified, perhaps the authors could focus on just these for their analyses.

This valid point constitutes one main challenge for manual cell annotation in whole organ imaging, as touched upon under the previous reviewer comment. The system employed here for neuronal annotation as proof of concept relies on an estimated neuron location relative to the sagittal plane, anterior-posterior axis and ventral-dorsal axis (CPV = centroposterior-ventral). In Figure 3-Figure Suppl.1 we have plotted mapped neurons using more narrow constraints than previously in Figure 3, in line with the reviewer’s comment for more focused analyses. However, in order to retackle these identities for further characterization and for building a reference retraceable naming system, our tools constitute only one building block and should be complemented by:

– A working model for cell stamping of known GRNs within a whole organ labeling in physiological recordings,

– A tissue registration approach of whole organ imaging for automatic cell mapping circumventing manual cell annotation and

– A higher number of Gal4 transgenic lines mapped alongside the 7 known identities of GR-expressing neurons, such as co-mapping GR lines with IR and Ppk Gal4 lines.

Also, considering that individual physiological characterizations for the few known GRNs of the larval TOG organ have been tackled previously for 5 out of 7 mapped neurons, the main aim of our study was rather towards an overview of the range of response types per organ, attempted for the first time in larval peripheral taste sensing, to our knowledge. We thus could identify an average cell response multimodality ratio per organ, cell-specific deactivation events and sucrose-evoked responses in the larval TOG. This informed the second part of this manuscript for sugar sensing in the larva and could similarly direct further dissections of larval taste coding.

4. The methods and criteria used for image acquisition and processing needs to be described better.

We appreciate this feedback and have clarified the methods description, all textual changes marked with track change.

– BehaviourThe authors' central thesis – peripheral sweet neurons>hard food; internal sweet neurons>soft food – hinges on the result that on hard food, despite the absence of 43a from the pharyngeal neurons, larvae still show a preference for sugar. Presumably larvae are not eating during the 2-5 minute window (4B) and so the brain 43a receptors cannot driving this preference for sugar. They reason, it must be peripheral neurons that are responsible for this. We have three issues with this central finding and their interpretations:1. Sugar or Texture? There's a possibility that mechanosensation (response to texture of the food) might be playing a role in this observation. The authors could demonstrate the specificity of this response by performing a dose response curve with sucrose (we recommend sticking to sucrose as it shows more robust PIs). So, we recommend that they perform the experimental conditions in 4B(bottom) with sucrose at different concentrations.

Thank you for your suggestion. We have modified the figure 4 to include 3 different sucrose concentrations for C2 on 1% versus 2.5% agarose, and have included in Figure 4-Supplemental1 two sucrose and fructose concentrations for *Gr43a* mutant/rescue genotypes.

Overall, these results indicate that sugar sensing in TOG neurons, specifically in C2, can detect sucrose on a hard substrate when food ingestion is impaired or slowed down. We propose that pharyngeal Gr43a neurons come into play mainly on ingestible substrate, at lower concentration, as also schematized in Figure 5. This hypothesis is supported by the expression of Gr43a receptors, which are present at the level of pharyngeal neurons, brain and other peripheral tissues such as in the foregut, but absent from the TOG external neurons (Kwon et al., 2011, Mishra et al., 2013).

2. The 43a null: If the peripheral neurons were driving this behaviour, the 43a null mutants should show at least a residual sugar preference, which they don't on either soft or hard food. It's possible that the authors are missing this in the low PIs of 4B. The above experiment should uncover this.

Indeed, as the reviewer correctly suggested, the above experiment shows residual preference of the Gr43a mutant to sucrose/fructose at higher concentrations.

3. Peripheral neurons on soft food only? What keeps the peripheral neurons from driving sugar preference on soft food in the absence of 43a? If the authors are suggesting that the peripheral sugar neurons are gated by both sugar and texture, they would need to demonstrate this through functional imaging. And in that case, activating the peripheral-GRNs in the brain-only 43a animal should induce sugar feeding on soft food.

We believe the above experiment equally addresses this point, as *Gr43a* mutant larvae do not show a complete defect on higher concentrations, including on 1% agarose. This indicates that sugar sensing via C2/TOG neurons contribute to output preference including during food ingestion but that it is overran by *Gr43a*-dependent pathways.

All in all, we do not believe C2/TOG neurons would be inactivated on soft food, but rather they play a role in a multicomponent system with different leverages for sugar sensing in the larva, as we are illustrating in Figure 5. Also worth noting is that the natural environment the fruit fly larva feeds on often implies the animal is immersed in a soft substrate, with a lesser need for clear taste dissociation between external and internal taste sensing.

In the individual reviews below, the authors will find more specific points that pertain to the broad points mentioned here. Could the authors please pay attention to them as they address these points?Reviewer #1:[…]My primary concern with this manuscript is the section dealing with the internal vs peripheral sensing of sugars. It's possible that some of my concerns arise out of the manner in which these results are reported and discussed, nevertheless, I will list them out here:– The preference indices in figure 4B seem very low: 0.2. At such low PIs I wonder if the authors might mistakenly interpret a lower π for a 'no-preference'.

Indeed, the 100mM Fructose concentration tests yield low preferences in the Gr43a mutant. We have therefore, at the reviewer’s suggestion, included higher tastant concentration in what is now Figure 4 – Figure Suppl.1.

In general I did not understand the reasons for using the various sugars at differing concentrations in this section.

To briefly describe our reasoning:

1. We have initially focused on sucrose 500mM because this was previously employed in other larval taste studies and in the whole organ imaging in the first part of the manuscript, which informed the behavioral testing in the second part.

2. Then, having formulated our hypothesis that larvae can sense sucrose despite not being able to readily ingest the food, we wanted to connect our observations with the previous reports for Gr43a sensing. As Mishra et al., 2013, have mainly described Gr43a sensing on fructose 100mM at 1% agarose concentration, we have added this sugar to our analysis to reproduce observations from this study and to compare between agarose concentrations.

3. Other sucrose and fructose concentrations have been added at the reviewer’s suggestion showing a lesser role of Gr43a at higher concentrations and also, as expected from literature reports, lesser to sucrose and more specific to fructose.

– The whole animal 43a mutant on both hard and soft food shows a complete inability to detect sugar. If this is true, then it suggests that the pheripheral neurons are NOT involved in sugar sensing. Do the authors agree with this interpretation?

At the reviewer’s suggestion we added behavior data in Figure 4-Figure Suppl.1 which show *Gr43a* mutant does not yield a complete defect for fructose 500mM preference, and neither is this the case for sucrose 100mM or 500mM.

However, we were generally not taken aback by an inability of the *Gr43a* mutant in sugar sensing, be it on soft or hard substrate. Since *Gr43a* is the main internal fructose sensor in *Drosophila*, expressed not only in the DPS, but also in the brain and in the proventriculus of the gastrointestinal system, we propose that a null mutant presents with a more systemic defect, which may be due to metabolic signaling, as shown for hemolymph carbohydrates influencing larval feeding behavior though insulin signaling and SLC5A11 (Ugrankar et al., 2018) or through the inability to gauge the state of satiety. Sugar sensing in the adult also consists of a multicomponent sensing system, with brad sugar sensing receptors on the proboscis distinct from Gr43a expressed internally that acts both as internal nutrient sensor and as feeding suppresor in satiated animals, while other receptors also carry roles in limiting sugar intake (Miyamoto et al., 2012; Joseph et al., 2012).

Altogether, we propose a model whereby C2 / TOG neurons act to signal the presence of sugar, but Gr43a expressed elsewhere can outweigh the behavioral output, either directly triggered by food ingestion, but also potentially indirectly through internal signaling (Figure 5B). This model for larval sugar-sensing mechanism aligns with, and complements, reports in the adult.

– The brain-only 43a mutant data makes the whole animal mutant data more difficult to interpret. If the difference between soft and hard food is a difference in time taken to ingest the food (as shown in Figure 4B), then there should be no difference in the trends of sugar responsiveness except in the axis of time. I have not understood why the authors think that on soft food the peripheral GRNs don't compensate for the lack of pharynx-43a to drive sugar feeding. Can the authors please comment on this?

Thank you. These are very interesting points.

Firstly, does the difference between soft and hard food preference for sugar resume to only time axis? We believe this is not the case because behaviors driven by C2 or Gr43a-neurons seem to not be merely additive in output but rather *Gr43a* internally guided preference would overrun, as proposed and schematized in Figure 5 according to our current understanding.

If Gr43a-neuron sensing outweighs sugar detection via C2 as we believe to be the case, this could also explain why peripheral GRNs don’t compensate for the lack of Gr43a. Gr43a is also expressed in other peripheral tissues such as the gut and the brain, acting as an internal nutrient sensor in both the larva and the adult (Miyamoto and Amrein 2013, Miyamoto et al., 2012, Mishra et al., 2013). Additionally, Gr43a seems to be more specifically tuned to fructose sensing, as *Gr43a* mutant is less impacting on the preference to sucrose (disaccharide of fructose and glucose), which is likely better compensated for in lack of *Gr43a* by external neurons (Figure 4 B). It’s worth mentioning that in the adult fly Gr43a receptor in the pharynx is also more specifically tuned for fructose sensing while external GR-expressing sweet sensing neurons are more broadly detecting sugars.

Reviewer #2:Maier et al., investigates neural basis of sugar perception/coding in *Drosophila* larvae using in vivo calcium imaging and behavior. Authors developed a volumetric imaging method to capture taste responses in all of the TOG neurons in larvae. The whole organ imaging method developed here will be a very useful technique for *Drosophila* larval chemosensory imaging. However, the imaging method and data analysis pipeline to identify cellular identity is not clearly explained in the manuscript. It is important to provide a metric for how effective and reliable cellular identity classifications are and the variations among individual larvae. The volumetric imaging results indicate that TOG neurons can respond to one modality or multiple modalities. The neural response dynamics are also variable. Some neurons show ON responses while some show OFF-responses or inhibition upon tastant stimulation. Furthermore, the authors test the function of TOG neurons in regulating sugar preference and found that the C2- TOG neuron is important for discriminating sugar when food is more viscous and harder to ingest. Overall, I think the volumetric imaging method is promising however, the imaging method and data analysis pipeline is not clearly explained and no metric has been provided to show how effective 3D cellular identity analysis is. The behavioral experiments regarding C2 neurons are not consistent with the previously published functional imaging analysis from the same group. Further functional imaging experiments are necessary to support the claims in the paper.1. The whole organ imaging method developed here will be a very useful technique for *Drosophila* larval chemosensory imaging. However, the details of image capture and processing is somewhat vaguely explained in the manuscript. Multiple details and metrics are missing about how imaging is done. What is the laser power used in these experiments? What is the frame rate for each volume scanned? What is the size of each volume, field of view etc. The movement and the drift correction for calcium imaging makes sense, however more detailed explanation about how they process imaging data is necessary. For example, in Figure Supplement 1, the authors are correcting GFP channels using the position information from the RFP channel and their method is able to handle the xy-plane. However, it is not clear how they handle z-drift. Did the post-processing also apply for the movement and drift correction in the Z-axis? What is the error rate for x-y or z correction? It would be good to know how accurate their methods are in identifying the same cell identities across animals. The authors need to expand the methods section and explain clearly how volumetric imaging and data processing was done in the paper. This is key for accurate interpretation of cell identification and the calcium imaging results.

Thank you for your observation, we now added to the description in the Methods section *Microscopy settings*, lines 619-698.

3D drift correction explanations and corresponding reference for the drift correction plugin employed are included in Methods and we added additional specifications under Data processing and analysis of whole-organ physiological recordings (from line 744)*.*

2. Is figure 2B and 2C showing the percent responsive cells recorded across multiple animals or the percent responsive cells within an animal? I think it is important to clarify this point.

Thank you for pointing this out, we recognize the importance for this elucidation and have attempted to describe the data more clearly. Figure 2 shows data pooled across multiple recordings, responding neurons were pooled together and the multimodality percentage was thus calculated over the entire dataset. Figure 2 B and C represent all responsive cells from organs stimulated with mixtures of tastants, while Figure 2 D and D’ represent all responsive cells from organs stimulated with single tastants.

In Figure 2-figure supplement 1 we have now also illustrated data per animal, as alternative representation of Figure 2 D’. This newly added figure shows the unimodality vs multimodality percentage calculated within each of the 15-recorded organs, giving a total average of 67% unimodality vs 33% multimodality. These numbers are very closely mirroring the 68%/32% fraction from Fig2D’ for all pooled responses.

What is the variance between individuals? I would like to understand how reliable cell identification and functional imaging analysis is across animals. This can be achieved if authors show data per animal rather than lumping all the data across the animals together. Another reason to visualize individual animal responses is to investigate whether the order of tastant stimulation is a critical factor for larval taste perception: For a neuron that can respond to bitter and sugar, would the sequence of stimulation matter? Will the neuron change activity if it was stimulated first with a bitter compound then a sugar compound? What about the opposite? I strongly encourage authors to show more traces for individual animals across trials with different tastant stimulations as done in Figure 2D. This will help the reader understand what is the variation in taste responses across animals and it will also increase confidence for the consistency of authors' volumetric imaging method.

Thank you, we agree with your observations and with the validity of the raised points.

We have included more traces in Figure 2 D for 2 separate representations of traces per organ. We have also added in Figure 2-Figure Suppl1 an alternative representation per each recorded organ of the multimodality computations done on pooled data in Figure 2D’ as mentioned under the previous reviewer comment. This will give a feel of how variable the captured neuronal responses were across organs, without taking into account cell identity mapping.

Regarding order of tastants: this aspect was incorporated in the experimental design through a randomized order of stimulation. We considered through this approach an eventual effect caused by the order of application would be averaged out as noise. To exemplify this, we have now added as Figure 2-Supplement 2 single neuron calcium imaging recordings in C7 (TOG neuron activated by both sugar and bitter modalities – van Giesen, 2016a). The order of presentation of two substances, sucrose and denatonium, yields no significant difference in sucrose responses recorded in this neuron.

We also replaced Figure 3 to add illustrative traces for some of the mapped neurons. To your comment we would like to add that ease in identifying cell identities equally depends on the nature of recorded tissue. For TOG cells we have observed relatively high variability of cell localization in live imaging as well as in immunostainings due to a supple architecture of the organ. On the other hand, we have found the DOG organ, which does not make the subject of this work, to be more compact with fewer variables in cell position. Similarly, the adult proboscis has a fairly rigid conformation and hair bristles that aid for referencing in cell identification. All in all, consistency of our imaging tools in the TOG can be improved through future efforts by using complementary methods for cell labeling and/or tissue registration.

3. In figure 3A, the authors show different calcium dynamics upon tastant stimulation in different TOG neurons. They have observed, off responses and inhibition of neural activity in certain neurons upon stimulation. I would like to see what is the percentage of OFF-responses and inhibition observed in these experiments. Are these events frequent or rare? Are they seen in all animals or some of the animals used in the calcium recordings? It is important to understand whether these are biologically relevant neural responses or artifacts seen in some of the animals used in the experiments.

OFF-responses have been assigned when a neuron showed activation after the stimulation, and a deactivation response was considered when a neuron responded with a decrease in florescence during stimulation. It remains to be determined if the 2 types of events are related to distinct cellular mechanisms or if they are coupled, but it is often the case that a neuron responds with both deactivation and a subsequent OFF-response to a tastant.

In particular, such events have been systematically assigned to the CDL2 neuron as presented in Figure 3-figure supplemental 1 B. Citric acid, employed for stimulation in all 15 organs subjected to individual tastants, yielded a deactivation in an approximated cell location named by us CDL2 in 9 out of 15 recorded organs (plotted in Figure 3-Figure Suppl.1 B). An OFF-response to citric acid in the same neuron was actually captured in 11 out of 15 organs. The two numbers don’t coincide because the amplitude for deactivation or OFF-response didn’t always simultaneously meet the criteria of 20% threshold, but all in all we consider a deactivation/OFF-response to citric acid in CDL2 neuron to be more than merely an artifact.

Other tastants than citric acid have elicited OFF-responses but less robustly, especially in CDL2 but also in other neuronal locations. Generally, we believe such events are neuron and tastant/concentration specific and that a larger range of tested tastants and concentrations would reveal further such neuron-deactivation pairings. However, CDL2 associated most deactivation signals in our dataset (included in source data supplementary files), indicating this neuron could comport distinctive tuning properties.

4. In Figure 3, authors replot the results from Figure2 this time by identifying cellular identity using anatomical position. This is something similar to what I was trying to get to in my previous comment (see comment 2). But even in plots shown in Figure 3C, it is hard to understand how successful authors' methods are in classifying each neuron type in each animal. For example, how many larvae in total have been tested in Figure 3C and in what percentage of the animals they were able to identify each neuron type and what is the variation in neural responses among each animal. I think this is a very critical point that needs to be clarified in the paper variation of taste responses among animals and a metric to show how successful cellular identification is across all the animals tested. If my understanding is correct, the total number of tested organs/animals is 15, however the data plotted in Figure 3C does not correspond to this number. Please explain what is the success rate to define a cell type across animals.

We believe the whole-organ approach described here can serve for future characterizations with more precise neuronal annotation results by using complementary tissue registration or double neuronal labeling techniques. Within the purpose for this study we have manually mapped cells for which a tastant was found to yield a response in at least 3 different instances. Therefore, for some data plotted in the Figure 3-Figure Suppl1 (previously Figure 3) the N was 3, while a given tastant was presented in at least 7 recordings, except for sucrose and citric acid that were used for stimulation in all 15 organs. We have now restricted the threshold to N>=4, being at least half of stimulated organs with a given tastant, except for bitter stimulations for which N>=3 as these responses were more fade and more difficult to map. This threshold does not necessarily mean a given neuron only responded in 3-4 organs out of 7-8 stimulated with a given tastant but it can also mean one of the following:

1. The response was not picked up above 20% DF/F0 amplitude or

2. the fluorescent trace was filtered out due to unstable baseline or high variance of signal before stimulation or

3. The response could not be unambiguously assigned to the given cell location or

4. the experimenter was not confident about the cell identity even if a response to the given tastant was picked up.

Therefore we would like to propose Figure 3-Figure Suppl.1 should be interpreted as illustration of cell tuning types that can be picked up in the TOG, rather than a complete cell mapping of the organ.

Nevertheless, the reproducible mono/multimodality percentages (illustrated per organ in Figure 2-Figure Suppl.1) and the captured deactivation/OFF-responses add to our understanding of the taste sensing range in the larval external organ.

5. The authors mention that the only known sugar receptor gene Gr43a is not expressed in TOG but TOG neurons can respond to sugars. This is a puzzling finding. Is the expression analysis of Gr43a receptor based on transgenic reporters or single cell sequencing? In other words, is there other evidence that TOG does not express any known sugar receptors in larvae?

Yes, indeed, thus far a sugar receptor expression in the larval TOG has not been reported in previous efforts for transgenic lines mapping. Except for a newly identified appetitive taste sensing to ribonucleosides via Gr28a (Mishra et al., 2018) and to certain amino acids via IRs (Croset et al., 2016), not much has been described for attractive behaviors attributed to neurons/receptors in the larval TOG.

Known sugar receptors (Gr5a, Gr64e-f) rely on description in the adult fly, but these have not been detected in the larval TOG. Conversely, certain Gr receptors are larval specific and incompletely characterized and therefore might cover sugar sensing at this developmental stage.

According to studies using transgenic reporters, larval Gr43a – similar to adult Gr43a – is expressed in internal tissues (pharyngeal taste neurons, gut, brain) but not in external sensing neurons.

We are presenting an enticing opportunity to investigate novel sugar receptors based on these results.

6. In figure 4, authors performed behavioral experiments to test the function TOG sugar sensing in larvae. Inhibiting one of the TOG neurons, C2 causes defects in sucrose preference when agarose ss 2.5 % but not 1%. They claim the external sugar sensing via TOG sensory neurons is critical when food is not soft enough to ingest. These are interesting hypothesis however, in their previous study, authors have shown C2 neurons respond weakly to 500mM sucrose but not to any other sugar. However, in this manuscript the stronger behavior phenotype is seen in 250mM fructose preference upon silencing of C2 neurons. It is not clearly to me what is the mechanism here. Please explain.

Indeed, we found C2 to respond physiologically to fructose 100mM with higher amplitudes than to fructose 500mM (data not shown), therefore this neuron’s role in fructose detection might depend to a large extent on concentration. Given that sucrose is a disaccharide constituted by fructose and glucose, we would expect C2 to also respond at least partially to fructose. On the other hand, it seems the Gr43a role in sugar detection is more specific for fructose than for sucrose, as also reported by others (Miyamoto and Amrein 2013, Miyamoto et al., 2012) and as we show in Figure 4-Figure Suppl.1 B.

Since the phenotype yielded by C2 in sucrose preference is more pronounced, we chose to focus on this sugar and to not include fructose preference indexes or physiological activation responses in C2, for simplicity and overall readability of the data.

7. In general, I think it will help to use one nomenclature for the paper or provide a table matching C-X identity to cellular identities in Figure 3C and list which compound(s) each neuron responds to and indicate the type of response; ON, OFF or inhibition.

Thank you for your suggestion, a table is included in the supplementary information (corresponding source data excel table).

8. The manipulations for larval GRNs in the manuscript are done chronically by expressing TNT-E. This might impact the development and wiring of the larval chemosensory system and cause problems in sugar detection unrelated to GRN function. I think using a conditional inhibitor of neural activity such as shits or newly developed ontogenetic inhibitors, GtACRs, might be better to use here to show conditional inactivity of the neurons result in similar defects in sugar sensing.

This is an excellent suggestion and we now incorporated in Figure 4 D new data for C2 neuron behavior for temporally controlled silencing with GtACR1. This experiment reproduces a defect to sucrose 500mM on 2.5% substrate, as observed with TNTE silencing.

Reviewer #3:*Drosophila* larvae have been put forth as a good model to study taste because of the simplicity of the system (in terms of the number of organs and neurons) and efficient behavior assays. However, an understanding of taste coding in the larval system lags behind that described for the adult stage, in which most taste sensilla/neurons are accessible for recordings or imaging. Maier et al., seek to address this gap by developing a whole organ calcium imaging approach to assess responses of neurons in the larval terminal organ to sweet, bitter, salt, amino acid, and acid tastants. From imaging 15 organs, the authors describe responses to a set of sweet, bitter, salt, amino acid, and acid tastants, collectively represented in neurons that appear to be tuned to one, two or multiple categories of tastants.Although the new approach has promise, the current study makes a limited conceptual advance and the number of positive responses used to ascribe responses to neurons is small. Overlap of tastant sensitivities has now been described, and more rigorously so, in the adult stage and in the authors' own previous work focusing on genetically labeled subsets of taste neurons. The identification of responsiveness to sucrose, which may be somewhat unexpected given the absence of sweet Gustatory receptors (Grs) in larvae, has also been previously described for a few neurons, one of which is further studied in behavior assays.The imaging analysis is followed by investigation of the potential roles of sugar sensing neurons in the periphery. While this analysis provides further support for a previously identified peripheral neuron (C2), it does not convincingly parse contributions of internal and external neurons.Specific weaknesses are elaborated via points below and could be addressed either with additional experiments or textual modifications.Calcium imaging:– Neuron categories were determined by imaging responses to only a single concentration of each tastant; more careful dose response analysis may alter the conclusions. This possibility has not been considered.

Indeed, this is a valid observation, and we have in fact noticed an important difference in responses to different concentrations of the same tastants, in fact as also exemplified by distinct responding cells and divergent behaviors elicited by low and high salt concentrations. This possibility is also discussed in the *Discussion section*.

Given the high level of complexity we faced in our analysis, particularly as the cell identity mapping was approached manually in this study, dose response would exponentially add to the complexity rendering manual response annotations impossible, and therefore we have focused on unique concentrations. We judge our tools valuable for future efforts in elucidating this complexity in tuning using various tastant concentrations with complementary methodology for tissue registration or double labeling of neurons that would make room for computerized cell response annotations.

– There is no effort to reconcile the imaging results with predictions from previous studies. For example, the number of bitter-responsive neurons is 1-2 per larva. However, bitter-GAL4 lines have been mapped to many more neurons in the TO.

In effect, our approach comports limitations in cell identification, as mentioned above. However, physiological characterization of bitter neurons in the TOG is relatively limited. Namely, neurons described to comport aversive behavior in the TOG are more or less 3: C1 to a diversity of bitter tastants (Kim et al., 2016, Choi et al., 2020), C3 to quinine (Apostolopoulou et al., 2014) and C7 to denatonium and quinine (van Giesen 2016a) and these are also summarized in Figure 5A. Any other contributions to bitter taste in the larva is inferred based on transgenic expression of GRs characterized as bitter sensing in the adult but not physiologically associated to bitter sensing in the corresponding larval neurons. All considering, bitter responding neurons per organ identified in our study is conform in number to previous physiological characterizations.

– The correspondence between neurons named from previous GAL4 mapping experiments to those named by position in this study is not at all clear. A better comparison between the two could be presented.

As restated in several other places above, the manual cell identification based on position did not allow for a high confidence association with previously Gal4-mapped neurons. We believe the tools presented by our manuscript can serve future studies in bridging the gap by using double labeling of neurons.

–The criteria that were used to manually exclude false positive responses in calcium imaging experiments (Line 475-476) have not been described.

Thank you for raising this valid point, a description is added under the corresponding Methods section *Data processing and analysis of whole-organ physiological recordings*, from line 744.

– A response was ascribed to an identified neuron if the position was the same in >3 organs. This threshold seems low, considering that 7-8 organs were tested with each stimulus. In the table for Figure 3, for instance, several tastants have n=3. Is this from a higher number of tastant applications? If so, what was observed in the other cases – that a response could not be ascribed to the identified neuron or that there was no response at all?

In other cases (instances where a response could not be ascribed to a given cell) the response was either not picked up / not above the set threshold of 20% amplitude, or the responding cell could not be confidently associated to a given identity.

Therefore we would like to propose Figure 3-Suppl.1 should be interpreted as illustration of cell tuning types that can be picked up in the TOG, rather than a complete cell map of the organ. The TOG tissue is flexible and shows a fluid structure, much more than its DOG counterpart that is more compactly organized. Therefore we anticipate the above question will be addressed by other studies by using a computerized tissue registration.

Nevertheless, the reproducible mono/multimodality percentages (illustrated per organ in Figure 2-Suppl1), the sucrose responses in the TOG, as well as the captured deactivation/OFF-responses, all add to our understanding of the taste sensing range in the larval external organ.

Behavior:– There are no indications of statistically significant differences between the genotypes in Figure 4B; the legend only mentions that the preference of Gr43a mutants is not significantly different from zero. Finally, data in Figure 4C are compared with Mann-Whitney tests, but ANOVA should be used instead.

Thank you for your very useful comment; we have now replaced Mann-Whitney by ANOVA tests, except for specific comparisons where 2 genotypes instead of 3 were measured in Figure 4-Supplement1. All statistical comparisons are indicated in the Supporting information (source data excel files).

– Whether sucrose response of internal neurons is affected in Gr43a mutants has not been tested (relevant to Figure 4C).

Thank you for raising this point – we have tested and now included this experiment in Figure 4-Figure Suppl.1 B. In our hands the *Gr43a* mutant shows a mild defect in sucrose preference (Figure 4 – Figure Suppl.1 B), which is confirmed from observations in the adult system where Gr43a has been described as an internal nutrient sensor (Miyamoto and Amrein 2013, Miyamoto et al., 2012) more specifically tuned to fructose than to other sugars, including sucrose (disaccharide of fructose and glucose).

– In Figure 4B right, assuming that Gr43a mutant controls are not significantly different from zero, whereas Gr43a "brain only" are, I don't know how to interpret a potential role for Gr43a brain neurons in preference for 2.5% agarose food, at 2 minutes when the larvae are not consuming anything.

We have now included more data points and proposed a supporting model in Figure 5. Indeed, we suggest that even for immediate preference at 2 min on 2.5% in absence of food ingestion a Gr43a mutant can still weigh on the output behavior. This could be due to Gr43a expression in several internal tissues (pharynx, proventriculus, brain) and to its role as hemolymph fructose sensor. It has been shown that hemolymph carbohydrates influence larval feeding behavior though insulin signaling and SLC5A11 (Ugrankar et al., 2018), but it remains to be determined if and how does a Gr43a mutant correlate with metabolic pathways and possibly interfere with sugar preference output behavior.

Furthermore, the data added for Gr43a mutant behavior in Figure 4-Figure Suppl.1 shows a lesser defect at higher concentrations and lesser in sucrose preference than fructose, implying a possible systemic effect of Gr43a more specifically to fructose, as mentioned under reviewer’s previous comment.

– The calcium imaging analysis is at the cellular level, not looking into contributions of receptors. Manipulating cells rather than receptors for behavior experiments would allow for better comparison between the two aims of the study. If the Gr43a mutant is the only way to go, suggested experiments are:1. Test if the complementary brain ablated genotype (Gr43a internal taste only) yields the expected phenotypes on 1% and 2.5% agarose.

Thank you very much for your suggestions. We have now added the complementary brain silencing genotype by reproducing the corresponding crosses described in the original paper by Mishra et al., 2013. We tested therefore Gr43a-Gal4-ki; Cha-Gal80 x *UAS-TNT* (Gr43a brain receptor silenced) on 1% and 2.5% (Figure 4-Figure Suppl.1 C). In the reference study for this test, the brain ablated Gr43a larvae don’t show a defect compared to control but on our hands a deficient preference was calculated for immediate preference on 1% and no defect on 2.5% percent. This difference with the reference paper might be due to laboratory dissimilarities in experimental manipulation, in fly food differences linked to subliminal metabolic state variations between animals or perhaps due to distinct potency of the *UAS-TNT*E line employed. A lack of defect of brain *Gr43a* silencing suggests that internal expression from pharynx and gut tissues is sufficient for internal sugar sensing.

2. Test if peripheral sugar responses are affected in Gr43a mutants.

Thank you, we followed your suggestion and found that *Gr43a* mutants associate some defect in sucrose preference but lesser than to fructose on both 1% and 2.5% (Figure 4-Figure Suppl.1 B and B’). This confirms adult fly reports describing Gr43a to have the highest specificity for fructose among sugars (Miyamoto and Amrein 2013, Miyamoto et al., 2012), including compared to sucrose (disaccharide of fructose and glucose).